# The organic cation transporter 2 regulates dopamine D1 receptor signaling at the Golgi apparatus

**Natasha M Puri[1], Giovanna R Romano[1,2], Ting-Yu Lin[3], Quynh N Mai[3], Roshanak Irannejad[1,3]\***

[1]Department of Biochemistry & Biophysics, University of California, San Francisco, San Francisco, United States; [2]Biochemistry Department, Weill Cornell Medicine, New York, United States; [3]Cardiovascular Research Institute, University of California, San Francisco, San Francisco, United States

**Abstract** Dopamine is a key catecholamine in the brain and kidney, where it is involved in a number of physiological functions such as locomotion, cognition, emotion, endocrine regulation, and renal function. As a membrane-impermeant hormone and neurotransmitter, dopamine is thought to signal by binding and activating dopamine receptors, members of the G protein coupled receptor (GPCR) family, only on the plasma membrane. Here, using novel nanobody-based biosensors, we demonstrate for the first time that the dopamine D1 receptor (D1DR), the primary mediator of dopaminergic signaling in the brain and kidney, not only functions on the plasma membrane but becomes activated at the Golgi apparatus in the presence of its ligand. We present evidence that activation of the Golgi pool of D1DR is dependent on organic cation transporter 2 (OCT2), a dopamine transporter, providing an explanation for how the membrane-impermeant dopamine accesses subcellular pools of D1DR. We further demonstrate that dopamine activates Golgi-D1DR in murine striatal medium spiny neurons, and this activity depends on OCT2 function. We also introduce a new approach to selectively interrogate compartmentalized D1DR signaling by inhibiting Gαs coupling using a nanobody-based chemical recruitment system. Using this strategy, we show that Golgi-localized D1DRs regulate cAMP production and mediate local protein kinase A activation. Together, our data suggest that spatially compartmentalized signaling hubs are previously unappreciated regulatory aspects of D1DR signaling. Our data provide further evidence for the role of transporters in regulating subcellular GPCR activity.

**\*For correspondence:**
roshanak.irannejad@ucsf.edu

**Competing interest:** The authors declare that no competing interests exist.

## Editor's evaluation

This study uses a tour de force of biosensor constructs providing evidence that dopamine transport by OCT2 across the plasma membrane and also (presumably) into the Golgi activates GPCR signaling at the Golgi, leading to cAMP production and PKA activation. Thus, intracellularly compartmentalized signaling underlies aspects of dopamine D1 receptor signaling. The work will be of interest to scientists working on the biology of dopamine signaling.

## Introduction

Dopamine (DA) is a major hormone and neurotransmitter that regulates a wide range of physiological responses, including reward-motivated behavior, aversion, cognition, and motor control in the central nervous system (CNS) (*Di Chiara and Imperato, 1988*; *Salery et al., 2020*; *Sulzer, 2011*). DA also regulates physiological responses in non-CNS tissues such as sodium secretion in the kidney

(*Missale et al., 1998*). All known cellular actions of DA are mediated by DA receptors, members of the G protein coupled receptor (GPCR) superfamily. Several pathological conditions such as Parkinson's disease, schizophrenia, and addiction are related to dysregulation of the neuronal dopaminergic signaling pathway, while hypertension has been attributed to impaired renal dopaminergic signaling (*Felder et al., 1990*; *Klein et al., 2019*). DA receptor antagonists have been developed with the goal of blocking hallucinations and delusions that occur in schizophrenic patients, whereas DA receptor agonists are used to alleviate the motor deficits of Parkinson's disease (*Felder et al., 1990*; *Klein et al., 2019*; *Urs et al., 2014*).

In both the CNS and kidney, DA is produced locally. There are five subtypes of DA receptors, D1, D2, D3, D4, and D5, that are classified as D1-class receptors (D1 and D5) or D2-class receptors (D2, D3, and D4) (*Kebabian, 1978*; *Spano et al., 1978*). The D1-class receptors are primarily coupled to Gαs/olf proteins and stimulate the activity of adenylyl cyclase (AC), leading to the production of the second messenger cyclic AMP (cAMP) (*Beaulieu et al., 2015*). In contrast, the D2 class are associated with Gαi/o proteins and inhibit cAMP production (*Beaulieu et al., 2015*). D1 dopamine receptors (D1DRs) are highly expressed in the CNS where they underlie major brain functions such as locomotion, learning and memory, attention, impulse control, and sleep (*Missale et al., 1998*). D1DRs in the kidney regulate trafficking of sodium ATPase and transporters, thereby affecting renal function (*Honegger et al., 2006*; *Wiederkehr et al., 2001*).

Impermeable agonists such as DA have long been thought to activate D1DRs only at the plasma membrane. Like many GPCRs, removal of D1DRs from the cell surface by endocytosis has been described as a mechanism that attenuates cellular signaling (*Ariano et al., 1997*; *Bloch et al., 2003*; *Vickery and von Zastrow, 1999*). As such, efforts at modulating DA signaling as a therapeutic strategy for various pathophysiological conditions have only taken into consideration the consequences of signaling by plasma membrane-localized DA receptors (*Jin et al., 2003*; *Panchalingam and Undie, 2001*; *Undie et al., 1994*). However, evidence from the past decade suggests that for some GPCRs endocytosis might in fact activate a second phase of acute or prolonged Gαs-mediated cAMP response from the endosomes (*Calebiro and Koszegi, 2019*; *Calebiro et al., 2010*; *Feinstein et al., 2013*; *Ferrandon et al., 2009*; *Irannejad et al., 2013*; *Irannejad et al., 2015*; *Irannejad and von Zastrow, 2014*; *Kotowski et al., 2011*; *Lobingier and von Zastrow, 2019*; *Stoeber et al., 2018*; *Thomsen et al., 2018*). Recent studies further support this notion by providing evidence that cAMP generation by activated receptors at the endosome is necessary to regulate transcriptional responses that are distinct from those elicited by activation of the plasma membrane receptor pool (*Bowman et al., 2016*; *Godbole et al., 2017*; *Jean-Alphonse et al., 2014*; *Jensen et al., 2017*; *Peng et al., 2021*; *Tsvetanova and von Zastrow, 2014*).

Most of the receptors that have been shown to exhibit a second phase of signaling from internal compartments are primarily coupled to Gαs protein. cAMP diffusion is within the nanometer scale around phosphodiesterases at physiological conditions (*Agarwal et al., 2016*; *Anton et al., 2022*; *Bock et al., 2020*; *Saucerman et al., 2014*). Given this narrow range of diffusion, it has been difficult to explain how receptor activation solely on the plasma membrane results in the activation of downstream effectors at distant subcellular locations such as the endoplasmic reticulum, Golgi, and nucleus (*Agarwal et al., 2016*; *Richards et al., 2016*; *Saucerman et al., 2014*). As one explanation for this observation, we recently showed that activation of the Golgi-localized beta1 adrenergic receptors (β1AR) causes local production of cAMP by Golgi-localized Gαs protein. Importantly, we demonstrated that a catecholamine transporter facilitates the transport of epinephrine, a membrane-impermeant endogenous β1AR agonist, to the lumen of the Golgi to activate the Golgi pool of β1AR (*Irannejad et al., 2017*). The importance of generation of a local pool of cAMP by Golgi-localized β1AR was further supported by the finding that activated Golgi-β1ARs, but not activated plasma membrane-β1ARs, cause PLCε activation at the perinuclear/Golgi membrane, which mediates hypertrophic responses in cardiomyocytes (*Irannejad et al., 2017*; *Nash et al., 2019*).

Whether the need for local cAMP generation is unique to cell types or specific GPCRs is not well understood. The lack of cAMP mobility in cells becomes prominent in larger cells with higher membrane compartmentation that present physical barriers to cAMP diffusion. Considering the high degree of membrane compartmentation of neurons and proximal tubules of the kidney, the two main cell types that express D1DRs, we wondered whether D1DR signaling is also compartmentalized. Here, using a conformational-sensitive nanobody that recognizes activated D1DR, we show that the

preexisting pool of D1DR that is localized to the Golgi membrane is activated upon stimulation with extracellular DA. In addition to several cell lines, here we demonstrate that Golgi-localized D1DR signaling is also a feature of primary striatal medium spiny neurons (MSNs). The D1DR-expressing MSNs of striatum are well established to play significant roles in motivation, aversion, and reward (*Alburges et al., 1992*; *Di Chiara and Imperato, 1988*; *Kim et al., 2015*; *Nestler and Luscher, 2019*; *Romach et al., 1999*). Furthermore, we demonstrate that OCT2 facilitates the transport of DA to the Golgi-localized D1DR and regulates its local activity at the Golgi. We further show that OCT2 has a distinct expression pattern in the kidney and specific regions of the brain, including the MSNs, where D1DRs are endogenously expressed. Thus, our findings reveal that DA can activate D1DR signaling at the Golgi and point to a novel role for OCT2 as a factor that determines which cell types exhibit DA-mediated subcellular signaling.

## Results

### Nanobody-based conformational-sensitive biosensors detect active D1DR and Gs protein at subcellular membranes

We have previously shown that a single-domain camelid antibody, nanobody 80 (Nb80), originally developed to stabilize an active conformation of beta 2 adrenergic receptor (β2AR) for crystallography purposes (*Rasmussen et al., 2011*), can be repurposed as a conformational biosensor to detect activated β2AR and β1AR in living cells (*Irannejad et al., 2017*; *Irannejad et al., 2013*). Through directed evolution on Nb80, a high-affinity nanobody (Nb6B9) was generated that stabilizes the active conformation of epinephrine-bound β2AR (*Ring et al., 2013*). Given that β2AR/Nb6B9 binding sites are highly conserved among other aminergic receptors such as β1AR and D1DR (*Figure 1—figure supplement 1a*; *Rasmussen et al., 2011*), we reasoned that this nanobody could also be used as a conformational-sensitive biosensor to detect activated D1DR in real time and living cells (*Figure 1a*). In HeLa cells expressing Snap-tagged D1DR, Nb6B9 fused to GFP (Nb6B9-GFP) was diffuse throughout the cytoplasm (*Figure 1b*, 0 min). Upon stimulation of these cells with 10 µM DA, Nb6B9-GFP was rapidly recruited first to the plasma membrane and shortly after to the Golgi apparatus (*Figure 1b*, 2 min, *Figure 1—video 1*). Nb6B9-GFP recruitment to the plasma membrane and the Golgi was dose dependent starting at 10 and 100 nM DA stimulations, respectively (*Figure 1c*, *Figure 1—figure supplement 1b*). Similar Nb6B9-GFP dose-dependent recruitments were observed upon activation of β1AR at the plasma membrane and the Golgi (*Figure 1—figure supplement 1c and d*). Importantly, no Nb6B9 recruitment to any membrane was detected when delta opioid GPCRs, which lack sequence homology to Nb6B9 binding sites, were activated (*Figure 1—figure supplement 1e*), suggesting the specificity of this conformational biosensor. Together, these data suggest that the D1DR Golgi pool is activated in response to extracellular DA addition. Contrary to HeLa cells, treatment of D1DR-expressing HEK293 cells with 10 µM DA resulted in the recruitment of NB6B9-GFP to only the plasma membrane (*Figure 1b*, lower panel, *Figure 1d*, 2 min, *Figure 1—video 2*). By contrast, SKF81297, a selective membrane-permeant D1DR agonist, activated both the plasma membrane and the Golgi receptor pools in both HeLa and HEK293 cells (*Figure 1—figure supplement 2*, *Figure 1—videos 3 and 4*). In addition to the Golgi recruitment and consistent with a previous report (*Kotowski et al., 2011*), Nb6B9 was also found to colocalize with D1DR at the endosomes, at a later time after agonist addition, indicating an active pool of D1DR at endosomes (*Figure 1—video 2*). We further used mini-Gs protein, a more general biosensor for Gs-coupled GPCRs (*Wan et al., 2018*), to show that the active pool of D1DR at the plasma membrane, endosomes, and the Golgi can also be detected by mini-Gs recruitment to these locations. Although we do not have exact measurements for Nb6B9 and miniG$_s$ binding affinities to activated D1DR, these observations suggest that miniG$_s$ is more sensitive, allowing the detection of activated D1DR at the Golgi starting at 10 nM DA addition (*Figure 1—figure supplement 3a and b*).

To investigate whether activated D1DRs couple to G proteins to elicit a G-protein-mediated response at the Golgi, we took advantage of another nanobody-based biosensor, Nb37-GFP. We previously used Nb37-GFP to detect transiently active β1AR/Gs and β2AR/Gs complexes at the Golgi and endosomes, respectively (*Irannejad et al., 2017*; *Irannejad et al., 2013*). Nb37-GFP was recruited to the plasma membrane and the Golgi upon stimulation with DA, suggesting that the D1DR Golgi pool couples to G protein and activates it (*Figure 1—figure supplement 4b and c*). Together, these

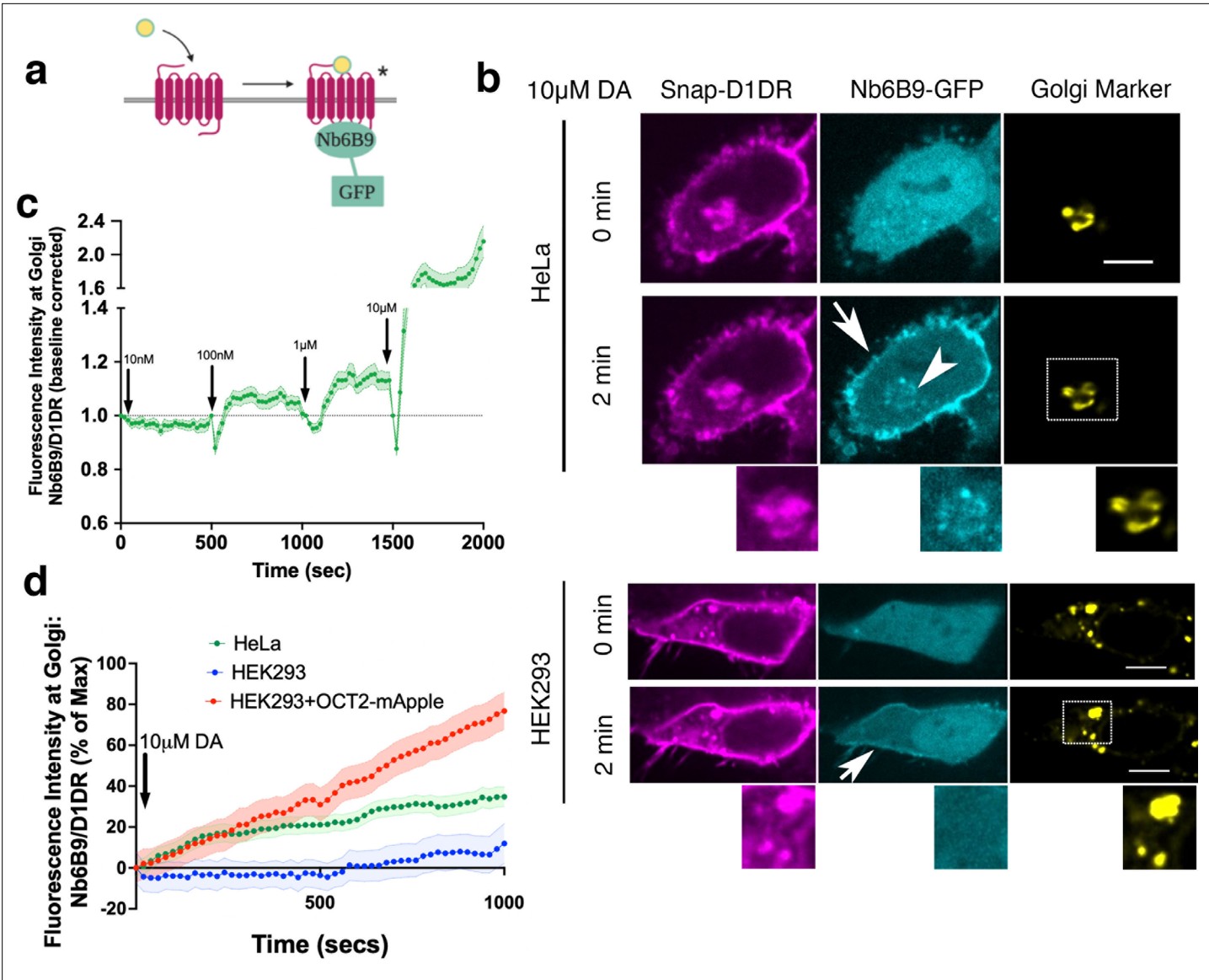

**Figure 1.** Conformational biosensor detects activated D1DR at the plasma membrane and the Golgi upon dopamine (DA) stimulation. (**a**) Nb6B9 binds to the receptor exclusively in its active conformation. We fused Nb6B9 to GFP and used it as a conformational biosensor for D1DR. (**b**) Confocal images of representative D1DR-expressing HeLa (top panel) and HEK293 cells (lower panel) with Nb6B9-GFP and GalT-mRFP expression before and after 10 μM DA addition. Stimulation with 10 μM DA results in recruitment of Nb6B9 to active D1DR at the plasma membrane and the Golgi in HeLa cells (n = 37 cells, Pearson's coefficient = 0.62, respectively, nine biological replicates); 10 μM DA treatment only activates plasma membrane-localized D1DR in HEK293 cells (n = 17 cells, Pearson's coefficient = 0.15, five biological replicates). Lower panels show zoomed images of insets for Snap-D1DR, Nb6BP, and the Golgi marker. Arrows indicate active D1DR at plasma membrane; arrowhead indicates active D1DR at Golgi membrane; Scale bar = 10 μm. (**c**) Quantification of D1DR activation at the Golgi in HeLa cells upon addition of increasing concentrations of DA; normalized fluorescence intensity of Nb6B9 at Golgi relative to Snap-tagged-D1DR at Golgi. Quantifications were baseline corrected after addition of each dose (n = 27 cells, four biological replicates). (**d**) Quantification of D1DR activity at Golgi in HeLa and HEK293 cells; normalized fluorescence intensity of Nb6B9 at Golgi relative to D1DR at Golgi labeled with Snap-tagged-D1DR.

The online version of this article includes the following video and figure supplement(s) for figure 1:

**Figure supplement 1.** Nb6B9 detects activated D1DR and β1AR at the plasma membrane and the Golgi in a dose-dependent manner, but not delta opioid receptors.

**Figure supplement 2.** Conformational biosensor detects activated D1DR at the plasma membrane and the Golgi upon SKF81297 stimulation.

**Figure supplement 3.** MiniGαs protein biosensor detects active D1DR at the plasma membrane and the Golgi.

**Figure supplement 4.** Conformational biosensors detect activated D1DR and Gs at the plasma membrane and the Golgi.

*Figure 1 continued on next page*

*Figure 1 continued*

**Figure 1—video 1.** Confocal image series of D1DR-expressing HeLa cells (magenta), Nb6B9-GFP (cyan), and the Golgi marker (yellow), incubated with 10 μM dopamine.

https://elifesciences.org/articles/75468/figures#fig1video1

**Figure 1—video 2.** Confocal image series of D1DR-expressing HEK293 cells (magenta), Nb6B9-GFP (cyan), and the Golgi marker (yellow), incubated with 10 μM dopamine.

https://elifesciences.org/articles/75468/figures#fig1video2

**Figure 1—video 3.** Confocal image series of D1DR-expressing HeLa cells (magenta), Nb6B9-GFP (cyan), and the Golgi marker (yellow), incubated with 10 μM SKF81927.

https://elifesciences.org/articles/75468/figures#fig1video3

**Figure 1—video 4.** Confocal image series of D1DR-expressing HEK293 cells (magenta), Nb6B9-GFP (cyan), and the Golgi marker (yellow), incubated with 10 μM SKF81927.

https://elifesciences.org/articles/75468/figures#fig1video4

findings suggest a distinct spatiotemporal regulation of D1DR signaling at the plasma membrane and the Golgi membranes.

## OCT2 facilitates the transport of dopamine to the Golgi-localized D1DR

These observations raised the key question of how DA, a hydrophilic/membrane-impermeant mono-amine, can access the Golgi pool of D1DR. The first clue came from the observation that DA activates Golgi-D1DR in HeLa cells but not HEK293 cells (*Figure 1b*, *Figure 1—videos 1 and 2*), whereas SKF81297, a hydrophobic/membrane-permeant agonist, activates the Golgi pool of D1DR in both cell types (*Figure 1—figure supplement 2*, *Figure 1—videos 3 and 4*). These distinct effects of DA and SKF81297 are not based on their differential potency for activating D1DR as they have comparative EC50 values in inducing cAMP responses (*Figure 2—figure supplement 1a*). Moreover, D1DR activation at the Golgi is not dependent on receptor internalization as inhibiting endocytosis by blocking dynamin did not block D1DR activation at the Golgi (*Figure 1—figure supplement 4a and c*). We previously found that a membrane transporter, organic cation transporters 3 (OCT3), facilitates epinephrine transport, resulting in activation of the Golgi-localized β1AR. OCT3 is a member of the solute carrier (SLC) family 22, uptake 2 transporters that are electrogenic and transport catecholamines in a bidirectional manner. Importantly, OCT transporters are localized on the plasma membrane and intracellular compartments, including nuclear envelope, thus they can transport catecholamines across the plasma membrane and across internal membrane compartments (*Gasser et al., 2017*; *Irannejad et al., 2017*). Therefore, we hypothesized that another OCT family transporter can similarly function in DA transport to allow for its delivery to the Golgi and for the activation of Golgi-localized D1DR pools.

There are three main OCTs that have largely overlapping distribution but distinct substrates (*Nies et al., 2011*; *Roth et al., 2012*; *Schomig et al., 2006*; *Taubert et al., 2007*). OCT3 facilitates the transport of epinephrine and norepinephrine (*Nies et al., 2011*). DA has been identified as a key endogenous substrate for another member of the SLC22A family, OCT2 (SLC22A2) (*Amphoux et al., 2006*; *Bednarczyk et al., 2003*; *Busch et al., 1998*; *Gründemann et al., 1998*; *Schomig et al., 2006*; *Taubert et al., 2007*). Therefore, we asked whether OCT2 has a role in transporting DA to the Golgi membranes. We found robust OCT2 protein expression in HeLa cells as measured by Western blotting, whereas expression in HEK293 cells was significantly lower (*Figure 2—figure supplement 1b*). In immunostaining experiments, we found OCT2 localization on both the plasma membrane and the Golgi in HeLa cells by using an OCT2-specific antibody. This immunostaining was abrogated in HeLa cells expressing *SLC22A2*-specific shRNAs but not those expressing the control, scrambled shRNA (*Figure 2—figure supplement 1d*). To test the role of OCT2 in DA transport, we first used corticosterone, an inhibitor that has been shown to broadly inhibit OCTs but more frequently used to inhibit OCT3 (*Gasser and Lowry, 2018*; *Koepsell, 2019*; *Nies et al., 2011*). We found that corticosterone did not inhibit DA-mediated D1DR activation at the Golgi in HeLa cells (*Figure 2b*, *Figure 2—figure supplement 2a*). We then used imipramine, which at lower concentrations is thought to selectively inhibit OCT2. At both 10 and 100 μM concentrations, imipramine inhibited DA-mediated D1DR activation at the Golgi (*Figure 2b*). By contrast, SKF81297, a membrane-permeant D1DR agonist that can diffuse across membranes and does not require facilitated transport, could still access and activate Golgi-D1DR in imipramine-treated cells (*Figure 2a–c*, *Figure 2—video 1*). Next, we overexpressed

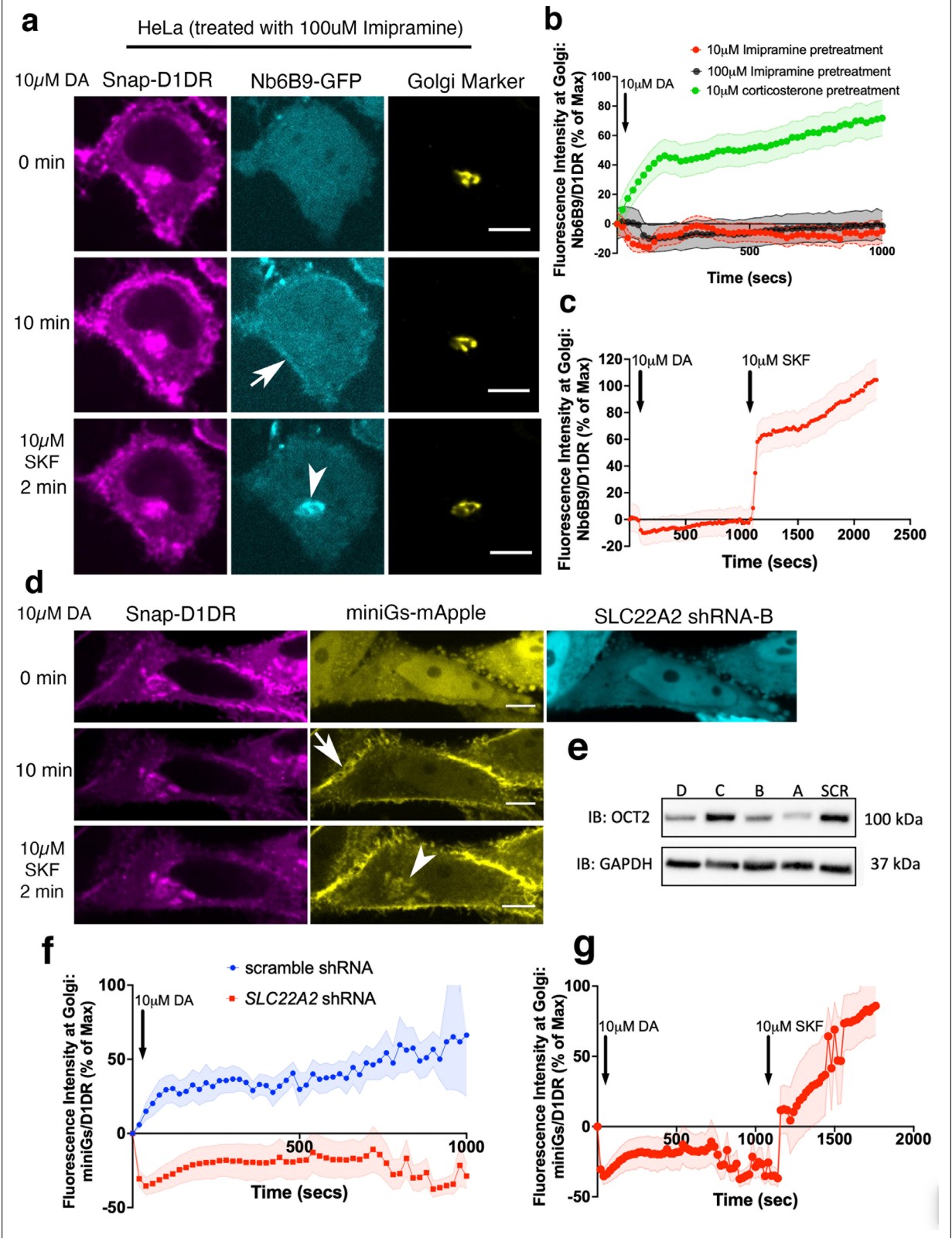

**Figure 2.** OCT2 facilitated dopamine (DA) transport to the Golgi-localized D1DR. (**a**) Representative HeLa cell expressing Snap-D1DR, Nb6B9-GFP, and GalT-mRFP expression pretreated with 100 µM imipramine for 15 min, before and after 10 µM DA addition. Inhibition of OCT2 blocks Golgi-localized D1DR activation but SKF81297 can still reach the Golgi membranes and activate D1DR Golgi pool (n = 30 cells, Pearson's coefficient = 0.2 and 0.68 after DA and SKF addition, respectively, three biological replicates). Arrow indicates active D1DR at plasma membrane; arrowhead indicates active D1DR at

*Figure 2 continued on next page*

*Figure 2 continued*

Golgi membrane; Scale bar = 10 µm. (**b**) Quantification of Nb6B9-GFP recruitment at Golgi upon 10 µM DA stimulation in HeLa cells pretreated with 10 and 100 µM imipramine, 10 µM corticosterone, and (**c**) after 10 µM SKF81297 addition; normalized fluorescence intensity of Nb6B9-GFP relative to Snap D1DR at Golgi (n = 30, three biological replicates). (**d**) Representative HeLa cell expressing Snap-D1DR, miniGs-mApple, and *SLC22A2* shRNA-B-GFP, before and after 10 µM DA addition. *SLC22A2* shRNA blocks Golgi-localized D1DR activation. 10 µM SKF81297 addition activates D1DR at the Golgi (n = 16 cells, Pearson's coefficient = –0.05 and 0.74 after DA and SKF addition, respectively, three biological replicates). (**e**) Detection of OCT2 expression in HeLa cells expressing different shRNAs by Western blot. (**f**) Quantification of D1DR activation at the Golgi in HeLa cells expressing scramble or *SLC22A2* shRNAs upon addition of 10 µM DA; normalized fluorescence intensity of Nb6B9 at Golgi relative to Snap-tagged-D1DR at Golgi. (**g**) Quantification of D1DR activation at the Golgi in HeLa cells expressing *SLC22A2* shRNA-B and D upon addition of 10 µM SKF81297.

The online version of this article includes the following video and figure supplement(s) for figure 2:

**Figure supplement 1.** OCT2 expression is required for facilitated transport of DA to the Golgi-localized D1DR.

**Figure supplement 2.** OCT2, but not OCT3, is required for DA-mediated D1DR activation at the Golgi.

**Figure 2—video 1.** Confocal image series of D1DR-expressing HeLa cells (magenta), Nb6B9-GFP (cyan), and Galt-mApple (yellow), pretreated with 100 µM imipramine and incubated with 10 µM dopamine and SKF81297.

https://elifesciences.org/articles/75468/figures#fig2video1

**Figure 2—video 2.** Confocal image series of D1DR-expressing HEK293 cells (magenta), Nb6B9-GFP (cyan), and OCT2-mApple (yellow), incubated with 10 µM dopamine.

https://elifesciences.org/articles/75468/figures#fig2video2

**Figure 2—video 3.** Confocal image series of D1DR-expressing HeLa cells (magenta), miniGs-mApple (yellow), expressing *SLC22A2* shRNA-B and incubated with 10 µM Dopamine for 20 min and then 10 µM SKF81297.

https://elifesciences.org/articles/75468/figures#fig2video3

OCT2-mApple in HEK293 cells and used Nb6B9-GFP to assess D1DR activation. By overexpressing OCT2-mApple in HEK293 cells, we found that Nb6B9-GFP could now be recruited to activated D1DR at the Golgi membranes (*Figure 1c*, *Figure 2—figure supplement 1c*, *Figure 2—video 2*). To further confirm the role of OCT2 in DA transport, we used two different *SLC22A2* shRNAs to decrease OCT2 expression (*Figure 2e*). We found that DA-mediated, but not that of SKF81297, D1DR activation at the Golgi was inhibited in HeLa cells expressing *SLC22A2* shRNA (*Figure 2d–g*, *Figure 2—video 3*). Control, scrambled shRNA did not inhibit DA-mediated D1DR activation at the Golgi (Figure-*Figure 2—figure supplement 2b*). Together, these results suggest that OCT2 facilitates the transport of DA to the Golgi lumen where it then activates D1DR at the Golgi membranes.

## Regulation of dopamine-mediated activation of the Golgi-localized D1DR in striatal MSNs by OCT2

To investigate the role of OCT2 in D1DR signaling in physiologically relevant cell types, we measured OCT2 expression patterns in cell types derived from the kidney and the brain, the two main organs where D1DRs are known to have function. Previous reports, some of which were dependent on RNA measurements, had suggested that OCT2 is robustly expressed in the striatum and cortex, where D1DR is known to express and have function (*Castro et al., 2013*; *Hallett et al., 2006*; *Tang and Bezprozvanny, 2004*), but at low levels in the hippocampus and substantia nigra, regions in which D1DR also has known functions (*Alburges et al., 1992*; *Arnsten et al., 1995*; *Busch et al., 1998*; *Double and Crocker, 1995*). By Western blotting using a validated antibody (*Figure 2e*), we similarly found significant OCT2 expression in the striatum and cortex, low expression in the hippocampus, and negligible expression in substantia nigra (*Figure 3—figure supplement 1a*).

To determine the role of OCT2 in regulating a distinct pool of D1DR signaling in neurons, we isolated primary murine striatal MSNs, where OCT2 is expressed at high levels (*Figure 3—figure supplement 1a*; *Bacq et al., 2012*; *Matsui et al., 2016*). Within the striatum, D1DR-expressing MSNs have been shown to play roles in DA-regulated processes such as motivation, aversion, and reward seeking (*Alburges et al., 1992*; *Di Chiara and Imperato, 1988*; *Kim et al., 2015*; *Nestler and Luscher, 2019*; *Romach et al., 1999*). We detected endogenous D1DR on both the plasma membrane and the Golgi membranes in MSNs using two different D1DR antibodies (*Figure 3—figure supplement 1b*, lower panel, *Figure 3—figure supplement 1c*). D1DR immunostaining was diminished when MSNs were immunostained in the presence of D1DR blocking peptide (*Figure 3—figure supplement 1c*). Using the same OCT2 antibody that was used in HeLa cells (*Figure 2—figure supplement 1d*), we

also showed that MSNs express OCT2 on both the plasma membrane and the Golgi (*Figure 3—figure supplement 1d*). Stimulating D1DR-expressing MSNs with DA resulted in the recruitment of Nb6B9-GFP to both the plasma membrane and the perinuclear regions (*Figure 3a and c*, *Figure 3—video 1*). The perinuclear region in MSN is indeed colocalized with the Golgi membranes markers (*Figure 3—figure supplement 1b*, top panel). Importantly, OCT2 inhibition resulted in the inhibition of DA-mediated Golgi-D1DR activation. By contrast, the membrane-permeant SKF81297 activated D1DR at the Golgi (*Figure 3b and c*, *Figure 3—video 2*). To demonstrate that D1DR can form a functional complex with G protein at the Golgi in MSNs, we took advantage of Nb37-GFP to detect transient D1DR/G protein coupling. Similar to what we have observed in HeLa cells, DA stimulation resulted in the recruitment of Nb37-GFP to the Golgi, suggesting that the D1DR Golgi pool is able to couple to G protein and activate it in MSNs (*Figure 3d*). These data demonstrate that Golgi-localized G protein signaling by D1DRs occurs in a physiologically relevant cell type and that this signaling requires OCT2. Moreover, as there are cell types that express D1DR but not OCT2, our findings suggest that OCT2 expression could determine which cell types exhibit both plasma membrane and Golgi-localized D1DR signaling under physiological conditions.

## Golgi and plasma membrane-localized D1DR both contribute to the cAMP response

Our data suggested that the plasma membrane and the Golgi pools of D1DR both couple to the Gs protein. In addition to its presence at the plasma membrane, AC has been reported to localize at the Golgi/perinuclear membranes (*Boivin et al., 2006*; *Cancino et al., 2014*). We therefore asked whether D1DR/Gs complexes at both the plasma membrane and the Golgi activate Gs-mediated cAMP responses. To address this question, we utilized a rapamycin dimerization system composed of FK506-binding protein (FKBP) and FKBP-rapamycin binding domain of FRAP (FRB) to rapidly induce recruitment of Nb6B9 to specific membrane compartments. This makes it possible to specifically block D1DR/Gs coupling at each distinctly localized pool. We have previously shown that βARs nanobody, Nb80, which binds to the same region as G protein (*Chung et al., 2011*; *Rasmussen et al., 2011*), blocks either the plasma membrane or the Golgi-β1AR-mediated cAMP responses when it is recruited locally to these compartments at high concentrations (*Irannejad et al., 2017*). This inhibition is likely due to steric occlusion of the Gαs protein. Using HEK293 cells expressing either FKBP at the plasma membrane or the Golgi with FRB fused to Nb6B9 (FRB-Nb6B9), we demonstrated that treatment with rapalog, a rapamycin analog, specifically targets Nb6B9 to either membrane (*Figure 4a–c*). Upon stimulation with membrane-permeant agonist SKF81297, Nb6B9 targeted to the plasma membrane disrupts plasma membrane-D1DR/G proteins coupling, while Golgi-D1DR is still able to elicit a cAMP response (*Figure 4d*). In turn, treatment with rapalog in cells expressing Golgi-targeted FKBP and FRB-Nb6B9 and subsequent stimulation with SKF81297 resulted in inhibition of the Golgi-D1DR pool (*Figure 4e*). Importantly, rapalog treatment alone had no effect on the overall cAMP production elicited by Forskolin, a direct activator of AC (*Figure 4f*). These data indicate that Golgi-localized D1DR is able to promote cAMP response.

## Local activation of PKA at the Golgi depends on selective activation of Golgi-localized D1DR

A key downstream effector sensed by cAMP is protein kinase A (PKA). PKA is a holoenzyme, consisting of two catalytic and two regulatory subunits (*Figure 5a*). There are two PKA types (types I and II) that are anchored to distinct subcellular locations through interactions with distinct A kinase anchoring proteins (*Soberg and Skålhegg, 2018*). PKA type II has been shown to localize to the perinuclear/Golgi membranes (*Nigg et al., 1985*). Binding of cAMP to the PKA regulatory subunit induces rapid dissociation and activation of the PKA catalytic subunit (*Figure 5a*; *Tillo et al., 2017*; *Walker-Gray et al., 2017*). To test whether cAMP generation by Golgi-localized D1DR/Gs complex results in the activation of PKA at the perinuclear/Golgi, we utilized a previously described HEK293T knock-in cell line expressing a split fluorescent protein, labeling native PKA catalytic subunit gene with GFP (PKAcat-GFP) (*Feng et al., 2017*; *Peng et al., 2021*). Stimulation of HEK293T PKAcat-GFP knock-in cell lines expressing D1DR with 10 nM SKF81297, a concentration that activates both pools of D1DR (*Figure 1—figure supplement 3a*), resulted in rapid dissociation of PKAcat-GFP from the perinuclear/Golgi membranes (*Figure 5b*, top panel). Quantification of these data shows that

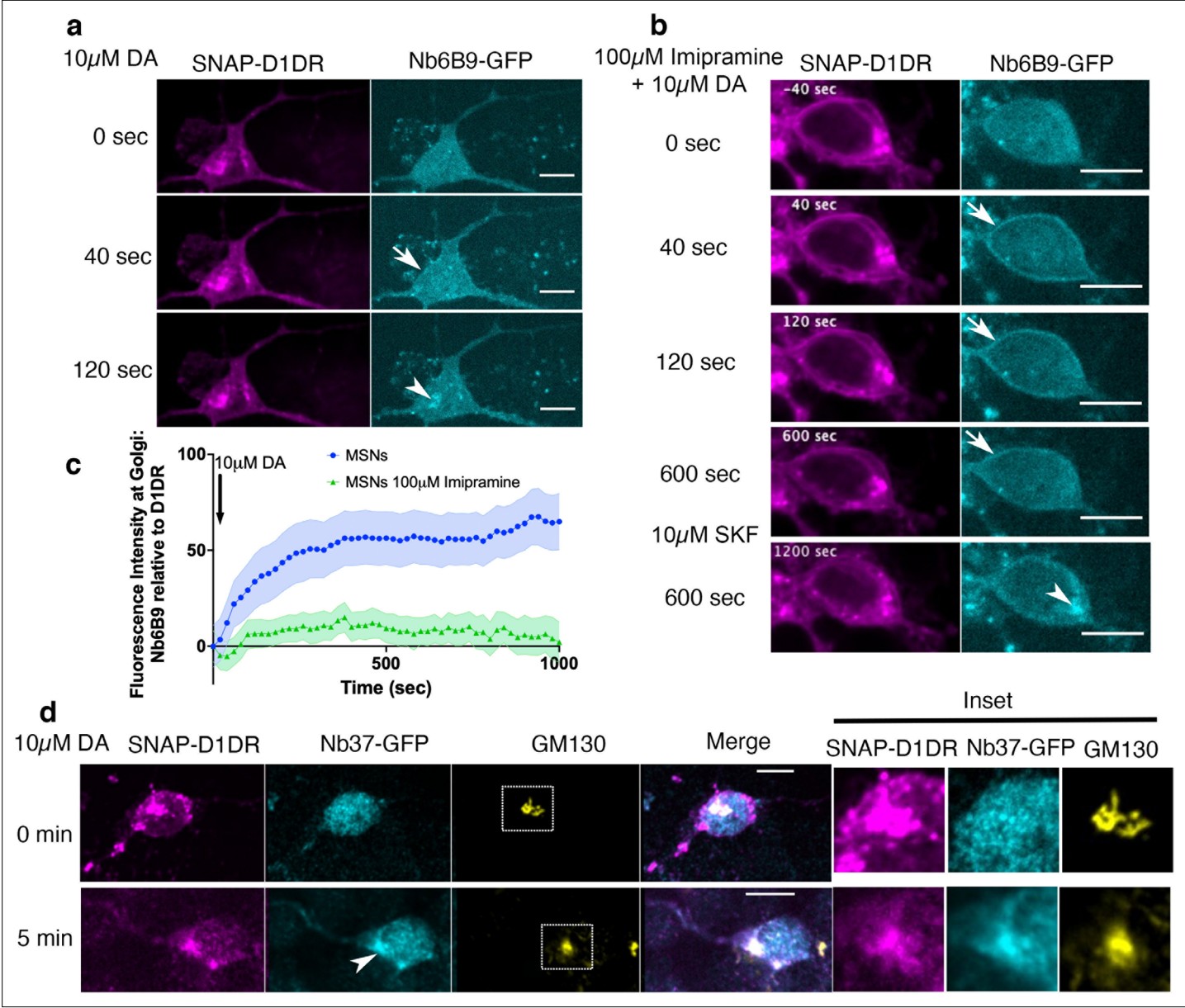

**Figure 3.** Regulation of dopamine (DA)-mediated activation of the Golgi-localized D1DR in striatal neurons by OCT2. (**a**) Representative medium spiny neurons (MSN) expressing Snap-D1DR and Nb6B9-GFP at indicated times after 10 μM DA addition. DA stimulates D1DR activation at the Golgi in MSNs (n = 22 cells, Pearson's coefficient = 0.67, six biological replicates). Arrow indicates active D1DR at plasma membrane; Arrowhead indicates active D1DR at Golgi membrane; Scale bar = 10 μm. (**b**) Representative MSN cell expressing Snap-D1DR and Nb6B9-GFP, pretreated with 100 μM imipramine for 15 min, before and after 10 μM DA addition. Inhibition of OCT2 blocks Golgi-localized D1DR activation at MSN n = 18 cells, Pearson's coefficient = 0.38, six biological replicates but SKF81297 can still reach the Golgi membranes and activate D1DR Golgi pool (n = 6 cells, Pearson's coefficient = 0.75, four biological replicates). Arrow indicates active D1DR at plasma membrane; Arrowhead indicates active D1DR at Golgi membrane; scale bar = 10 μm. (**c**) Quantification of Nb6B9-GFP recruitment at Golgi upon 10 μM DA stimulation in MSNs cells pretreated with OCT2 inhibitor; normalized fluorescence intensity of Nb6B9-GFP relative to Snap D1DR at Golgi (n = 12 and 7, respectively, five biological replicates). (**d**) Representative MSN expressing Snap-D1DR and Nb37-GFP before and after 10 μM DA addition. DA stimulates G protein activation at the Golgi in D1DR-expressing MSNs (n = 10 cells, Pearson's coefficient = 0.34 and 0.62 before and after 10 μM DA stimulation, six biological replicates). Arrowhead indicates active Gs at Golgi membrane; right panels show zoomed images of insets for Snap-D1DR, Nb37-GFP, and the Golgi marker (GM130). Scale bar = 10 μm.

The online version of this article includes the following video and figure supplement(s) for figure 3:

**Figure supplement 1.** Medium spiny neurons endogenously express OCT2 and D1DR at the plasma membrane and the Golgi.

**Figure 3—video 1.** Confocal image series of D1DR-expressing medium spiny neurons (MSNs) (magenta) and Nb6B9-GFP (cyan), incubated with 10 μM dopamine.

*Figure 3 continued on next page*

**Figure 3—video 2.** Confocal image series of D1DR-expressing medium spiny neurons (MSNs) (magenta) and Nb6B9-GFP (cyan), pretreated with 100 µM imipramine and incubated with 10 µM dopamine and SKF81297.

stimulation with SKF81297 results in sustained activation of PKA at the perinuclear/Golgi regions (*Figure 5c*, *Figure 5—figure supplement 1*). We then asked whether PKAcat dissociation is mediated by the activation of D1DR Golgi pool. Given that HEK293T express very low levels of OCT2 transporter (*Figure 2—figure supplement 1b*) and thus DA cannot be sufficiently transported to the Golgi membranes, we used DA to specifically activate the plasma membrane pool of D1DR. Importantly, stimulation of HEK293T PKAcat-GFP knock-in cells with 10 nM DA, a concentration with similar potency as SKF81297 (*Figure 2—figure supplement 1a*), did not promote PKAcat dissociation (*Figure 5b*, lower panel, and c, *Figure 5—figure supplement 1*). Together, these data indicate that

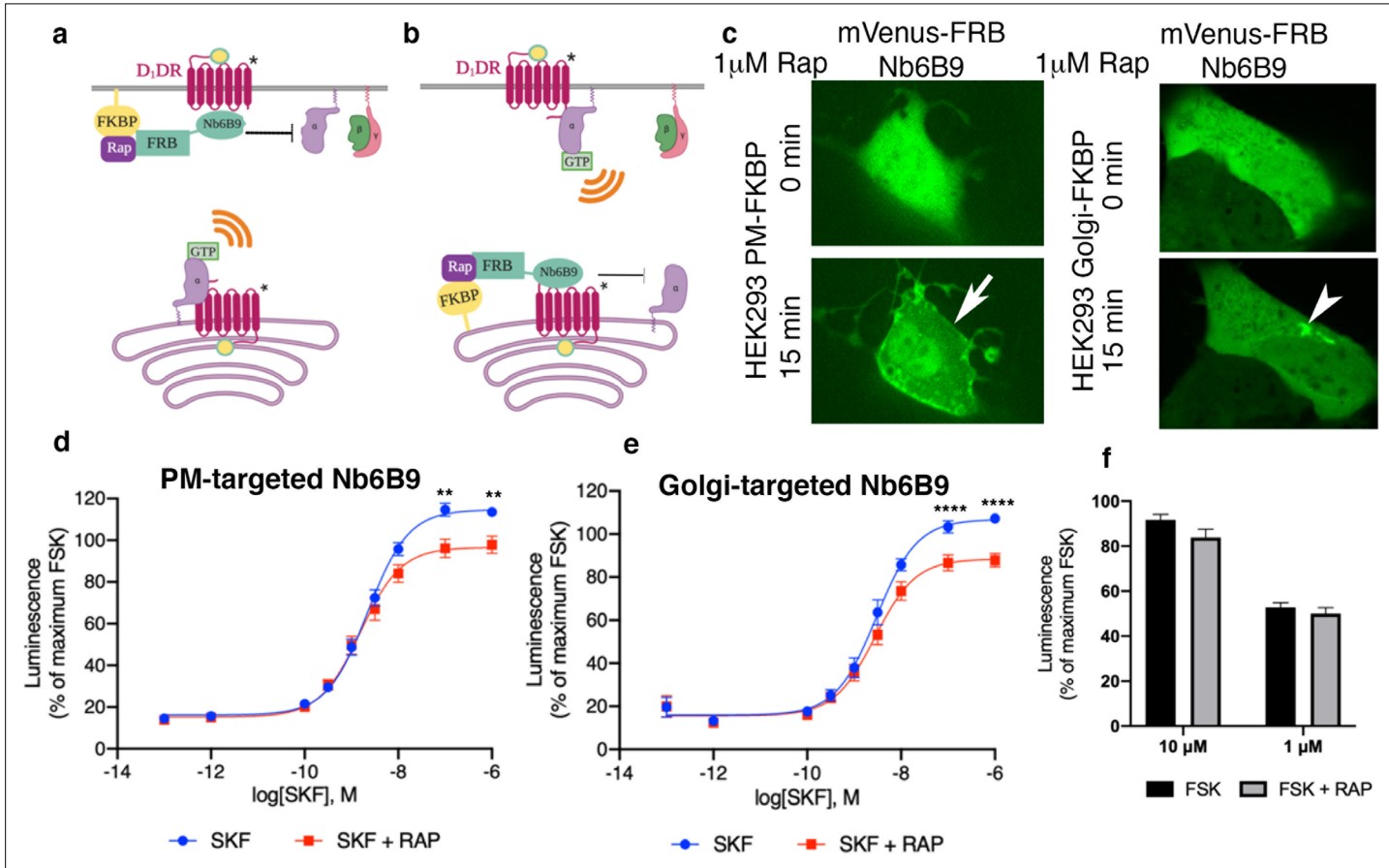

**Figure 4.** Both plasma membrane and Golgi-localized D1DR promote cAMP production. Model of blocking D1DR-Gs coupling at the plasma membrane (PM) (**a**) and the Golgi membrane (**b**) after recruitment of mVenus-FRB-Nb6B9. FKBP was targeted to either the PM (**a**) or the Golgi membrane (**b**), and its binding partner FRB-mVenus was fused to Nb6B9. Upon addition of rapalog (rapamycin analog), FKBP and FRB heterodimerize and sequester Nb6B9 to either membrane, disrupting G protein coupling to the receptor and thus blocking signaling from each respective location. (**c**) Representative confocal images of HEK293 cells expressing either PM or Golgi targeted FKBP showing mVenus-FRB Nb6B9 localization at indicated times after rapalog addition. Representative cells confirm inducible sequestration of Nb6B9 to either PM or Golgi. Arrow indicates PM; arrowhead indicates Golgi. (**d**) Forskolin-normalized D1DR-mediated cAMP response with and without rapalog pretreatment (1 µM, 15 min) and SKF81297 at indicated concentrations in HEK293-expressing PM-FKBP (mean ± SEM, n = 6 biological replicates, p-values of 0.0021 and 0.0015 at $10^{-7}$ and $10^{-6}$, respectively). (**e**) Forskolin-normalized D1DR-mediated cAMP response with and without rapalog pretreatment (1 µM, 15 min) and SKF81297 at indicated concentrations in HEK293-expressing Golgi-FKBP (mean ± SEM, n = 6 biological replicates, p-values of <0.0001 at $10^{-7}$ and $10^{-6}$). (**f**) Effect of 1 and 10 µM rapalog on forskolin-mediated cAMP response (n = 3 biological replicates).

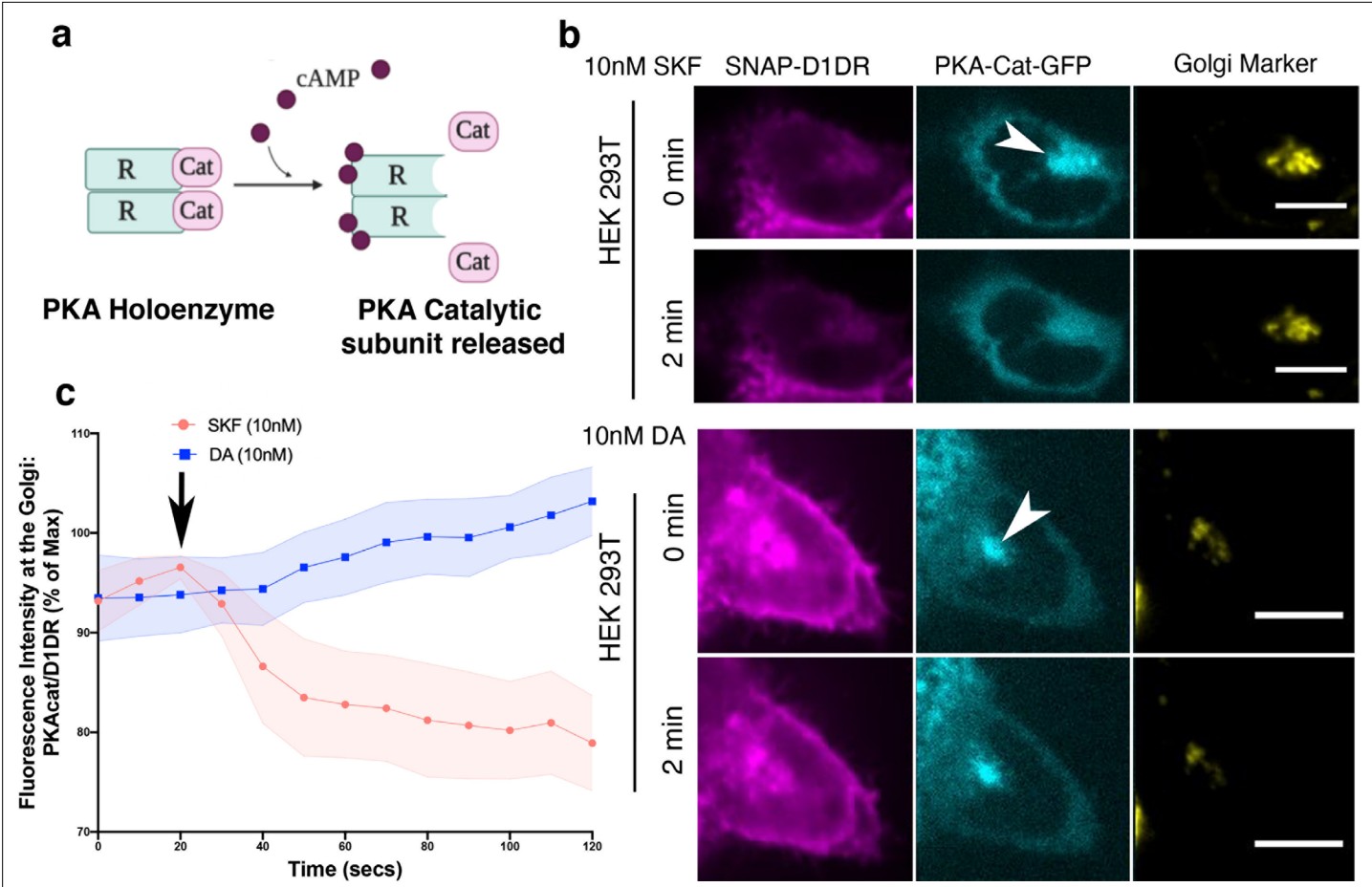

**Figure 5.** Golgi-localized protein kinase A (PKA) is activated by D1DR at the Golgi. (**a**) Model of PKA activation; cAMP binds PKA regulatory subunit (R), rendering PKA catalytic subunit (PKA-cat) dissociation. (**b**) Confocal images of representative D1DR-expressing HEK293 cells with endogenous PKA-cat-GFP and GalT-mRFP expression at indicated times after 10 nM SKF81927 (top panels; n = 11 cells, Pearson's coefficient 0.53, three biological replicates) or 10 nM dopamine (DA) (lower panels, n = 12 cells, Pearson's coefficient = 0.64, three biological replicates). Arrowhead indicates PKAcat at Golgi membrane; scale bar = 10 μm. (**c**) Normalized fluorescence intensity of PKAcat relative to Golgi-D1DR after treatment with 10 nM DA or 10 nM SKF81927.

The online version of this article includes the following figure supplement(s) for figure 5:

**Figure supplement 1.** Kinetics of PKAcat-GFP dissociation from the Golgi membrane upon activation of the D1DR Golgi pools.

activation of D1DR at the Golgi, but not the plasma membrane, results in local PKA activation at the perinuclear/Golgi regions.

## Dopamine uncaging triggers rapid activation of Golgi-localized D1DR and local PKA

To further investigate the role of Golgi-localized D1DR in activating PKA locally, we utilized a photosensitive caged DA that becomes uncaged upon blue light exposure (***Figure 6a***). Unlike DA, caged DA is hydrophobic and thus membrane permeant (***Castro et al., 2013***; ***Yapo et al., 2017***). To ensure that caged DA accumulates inside the cell and reaches the Golgi-localized D1DR, we incubated HEK293T PKAcat-GFP knock-in cells with 1 μM caged DA for 10 min in a dark incubator. Addition of caged DA to HEK293T PKAcat-GFP cells did not activate D1DR, as indicated by cytoplasmic localization of Nb6B9-mApple, confirming that DA is inactive in its caged form (***Figure 6b***, top panel). Upon stimulation of cells with blue light for 10 s, we observed D1DR activation at the Golgi, as detected by rapid Nb6B9-mApple recruitment to the Golgi membranes within seconds after blue light exposure (***Figure 6b and c***, ***Figure 6—video 1***). This was then followed by PKAcat-GFP dissociation from the perinuclear/Golgi regions as a result of cAMP production and PKA activation (***Figure 6b and c***,

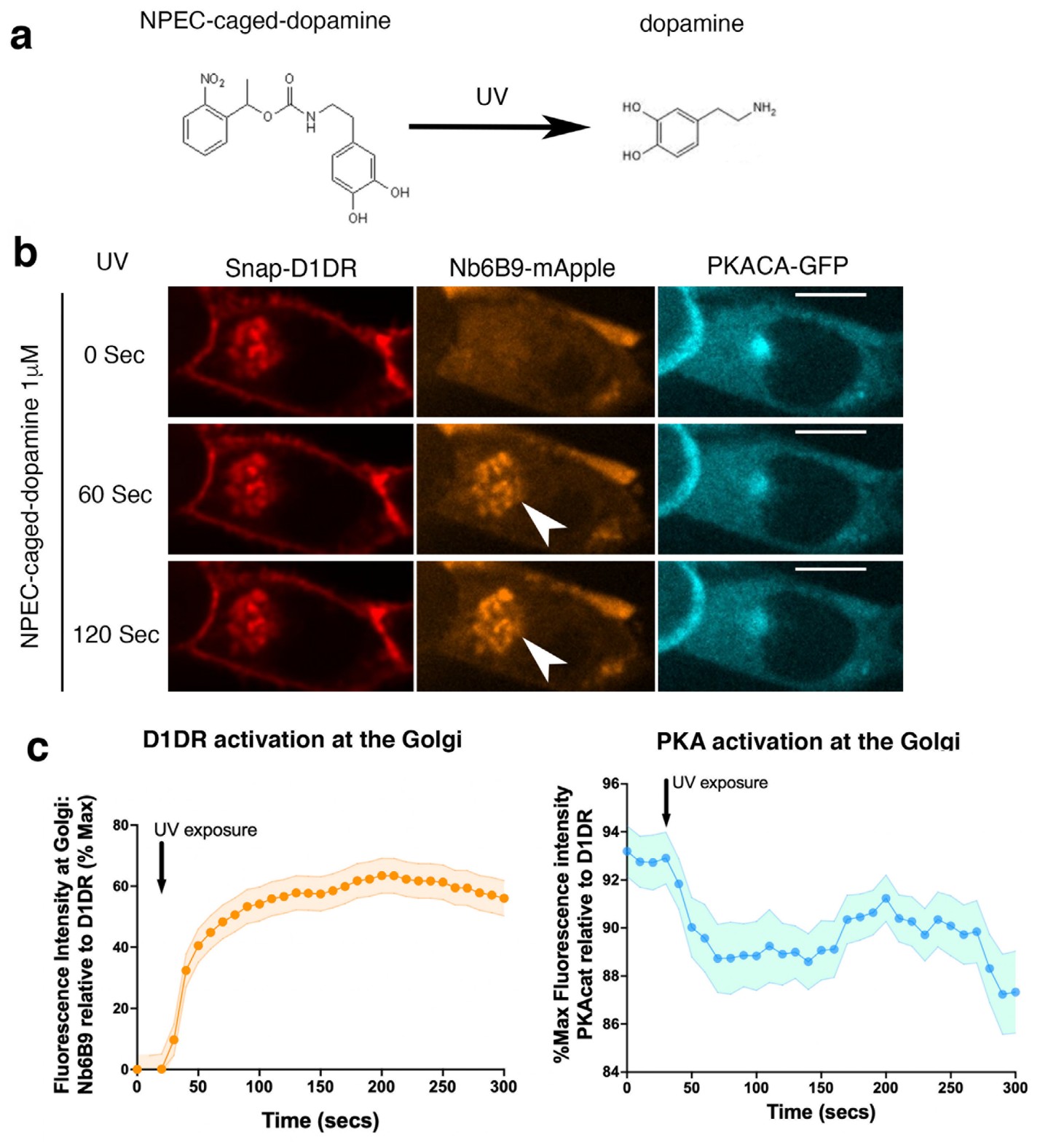

**Figure 6.** Rapid activation of Golgi-localized D1DR and protein kinase A (PKA) by photo-release of dopamine.
(**a**) Dopamine is uncaged from (N)–1-(2 nitrophenyl) ethylcarboxy-3,4-dihydroxyphenethylamine (NPEC) upon blue light (UV) exposure. (**b**) Confocal images of representative D1DR-expressing HEK293 cells with endogenous PKA-cat-GFP and Nb6B9-mApple expression, incubated with 1 μM NPEC-caged dopamine and at indicated times after blue light exposure (n = 46 cells, four biological replicates). Arrowhead indicates Nb6B9 recruitment to

*Figure 6 continued on next page*

*Figure 6 continued*

the Golgi membrane; scale bar = 10 μm. (**c**) Normalized fluorescence intensity of Nb6B9-mApple and PKAcat relative to Golgi-D1DR after blue light exposure.

The online version of this article includes the following video for figure 6:

**Figure 6—video 1.** Confocal image series of D1DR-expressing (magenta) HEK293 cells with endogenous PKA-cat-GFP (cyan) and Nb6B9-mApple (orange), incubated with 1 μM (N)–1-(2 nitrophenyl) ethylcarboxy-3,4-dihydroxyphenethylamine (NPEC)-caged dopamine.

https://elifesciences.org/articles/75468/figures#fig6video1

*Figure 6—video 1*). These data further support the notion that Golgi-localized D1DR activates PKA locally.

## Discussion

Our findings demonstrate for the first time that dopaminergic receptors can signal from the Golgi apparatus. We present evidence that DA, a hydrophilic catecholamine, can be transported to the Golgi membrane to reach the preexisting Golgi pool of D1DRs. This transport is facilitated by OCT2. The Golgi-D1DR comprises a functional signaling pool as it can activate Gαs and stimulate cAMP production. Moreover, we introduced a new approach to selectively interrogate compartmentalized D1DR signaling by inhibiting Gαs coupling using a nanobody-based chemical recruitment system. Finally, utilizing caged-DA, we showed that photo-release of DA at the Golgi upon rapid blue light exposure triggers D1DR-mediated cAMP production and local PKA activation.

As the signaling activities of D1DR have thus far been thought to be limited to the plasma membrane, substantial efforts have been focused on designing small-molecule agonists of D1DR to bias signaling towards a particular signaling pathway, without the consideration of spatial D1DR signaling (*Jin et al., 2003*; *Kuroiwa et al., 2008*; *Panchalingam and Undie, 2001*; *Undie et al., 1994*). Our findings on D1DR signaling from the Golgi membrane suggest location bias as an overlooked aspect of signaling specificity. This study demonstrates the important role of local generation of cAMP by GPCRs in controlling local PKA activation at specific subcellular compartments. It is well established that cAMP-mediated signaling specificity depends on the function of compartment-specific phosphodiesterases, enzymes that degrade cAMP, limiting the diffusion of this second messenger (*Agarwal et al., 2011*; *Buxton and Brunton, 1983*; *Gold et al., 2013*; *Musheshe et al., 2018*; *Steinberg and Brunton, 2001*; *Warrier et al., 2007*). Recent measurements of cAMP mobility suggest a nanometer-scale diffusion domain (*Anton et al., 2022*; *Bock et al., 2020*; *Saucerman et al., 2014*). The model where cAMP generation by plasma membrane-localized receptors propagates in a linear fashion to then control intracellular effectors of cAMP is inconsistent with the nanometer scale of cAMP diffusion range within the cell (*Agarwal et al., 2016*; *Saucerman et al., 2014*; *Zaccolo et al., 2021*). Thus, our data further provide evidence that PKA activation at a specific compartment requires GPCR activation locally in the vicinity of that compartment. Given that each subcellular membrane compartment has a distinct lipid environment (*Balla, 2013*), it is likely that PKA activation at each location will recruit a unique set of effectors and proteins and regulates distinct signaling and physiological outcomes. The importance of local generation of cAMP by Golgi-localized GPCR has been demonstrated for β1AR. Nash et al. have demonstrated that activated Golgi-β1ARs only, but not the plasma membrane pool, lead to PLCε activation at the perinuclear/Golgi membrane, which mediates hypertrophic responses in cardiomyocytes (*Irannejad et al., 2017*; *Nash et al., 2019*). Whether distinct signaling pathways are regulated by plasma membrane or Golgi-localized D1DR is not yet clear.

Published studies suggest that OCT2 is expressed in a number of tissues and cell types that also express D1DRs (*Arnsten et al., 1995*; *Busch et al., 1998*; *Double and Crocker, 1995*). Interestingly, however, there are D1DR-expressing cell types that do not express OCT2. For instance, we found that within the brain OCT2 is highly expressed in the striatum and moderately in the cortex. In contrast, OCT2 has little to no expression in the hippocampus or substantia nigra (*Figure 3—figure supplement 1a*; *Amphoux et al., 2006*; *Busch et al., 1998*). We also showed that OCT2 is expressed in MSNs in the striatum (*Figure 3—figure supplement 1d*). Consistent with this, we have further shown that DA activates Golgi pool of D1DR in MSNs (*Figure 3*). Moreover, we showed that activation of the D1DR at the Golgi, but not the plasma membrane, results in PKA activation at the perinuclear/Golgi region (*Figure 5*). This could potentially explain some of the distinct cAMP/PKA signaling patterns

that have been observed in different D1DR-expressing neurons. For instance, cAMP-dependent PKA responses have been shown to be sustained in striatal neurons compared to pyramidal cortical neurons (*Castro et al., 2013*). Therefore, we speculate that the expression pattern of OCT2 and activation of the Golgi-pool of D1DRs may be a determinant of which cell types and tissues exhibit sustained cAMP/PKA signaling.

D1DR-mediated cAMP signaling regulates major brain functions. Persistent DA stimulation and sustained receptor activation cause long-term changes in gene expression of neuronal plasticity-related genes, dendritic remodeling, and locomotor sensitization (*Gerfen et al., 1990*; *Le Moine and Bloch, 1995*; *Lüscher and Malenka, 2011*; *Zhang et al., 2006*). Many drugs of abuse increase the release of DA and elevate the firing rate of midbrain dopaminergic neurons in the striatum, particularly MSNs (*Di Chiara and Imperato, 1988*; *Lüscher and Malenka, 2011*; *Nestler and Luscher, 2019*). Whether this sustained receptor-mediated cAMP/PKA activation in MSNs is a consequence of D1DR activation at the Golgi is not clear, but strongly suggested by our findings. Understanding the contribution of D1DR subcellular signaling could potentially help with drug development for disorders where dopaminergic signaling is misregulated.

There are two major DA uptake transport mechanisms: (i) uptake 1 transporters that have high affinity for DA and are mostly localized in presynaptic neurons, and (ii) uptake 2 transporters that have low affinity but high capacity for DA and are expressed in various brain regions as well as different organs in the body (*Gründemann et al., 1998*; *Lin et al., 2011*; *Nies et al., 2011*; *Reith et al., 2006*; *Torres et al., 2003*). OCT2 belongs to the uptake 2 transporter family and has been previously thought to mainly function as an uptake transporter, helping with the clearance of extracellular DA and terminating DA-mediated signaling pathways (*Amphoux et al., 2006*; *Bednarczyk et al., 2003*; *Busch et al., 1998*; *Taubert et al., 2007*). Unlike uptake 1 transporters, uptake 2 transporters can transport catecholamines, including DA, across the membrane, in a bidirectional and electrogenic manner, and independent of $Na^+$ and $Cl^-$ transport (*Nies et al., 2011*; *Schomig et al., 2006*). Previous reports have demonstrated that OCTs, particularly OCT3, are localized on both the plasma membrane and subcellular membranes including the outer nuclear membranes near the Golgi (*Gasser, 2021*; *Gasser et al., 2017*). We showed that OCT2 is expressed on both the plasma membrane and the Golgi membranes in HeLa cells and MSNs (*Figure 2—figure supplement 1d*, *Figure 3—figure supplement 1d*). As OCT2 is a member of an electrogenic and bidirectional transporter, we speculate that the plasma membrane OCT2 facilitates the transport of DA from the extracellular environment to the cytoplasm and the intracellular-localized OCT2 might facilitate DA transport into the Golgi. Given that the resting membrane potential of inner nuclear membrane (~–100 mV) (*Burdakov et al., 2005*; *Matamala et al., 2021*; *Sanchez et al., 2018*) has been reported to be more negative relative to that of the cytoplasmic side of the plasma membrane (~–40 to –70 mV), it is plausible that, just as the transport of DA from the extracellular space into the cytoplasm by OCT2 takes advantage of the electrogenic gradient, a similar gradient allows for transport of DA from the cytoplasm across the nuclear envelope which is connected to the lumen of the Golgi membrane.

Accurate measurements of cytoplasmic DA in intact pre- or postsynaptic neurons have been challenging due to lack of sensitivity of most analytical methods and their effects on cell viability (*Chang et al., 2021*; *Olefirowicz and Ewing, 1990*; *Post and Sulzer, 2021*). However, given that DA is present at high millimolar concentrations within the synaptic vesicles (*Omiatek et al., 2013*; *Zhang et al., 2020*), it is likely that rapid uptake of DA post release will result in high cytoplasmic DA concentrations. With DA as the substrate, Km measurements ranging from 2 to 46 µM have been reported for OCT2 transporters (*Amphoux et al., 2006*; *Gasser, 2021*; *Gründemann et al., 1998*; *Schomig et al., 2006*). Thus, as a low-affinity but high-capacity transporter, subcellular OCT2 is likely to encounter high concentrations of cytoplasmic DA under physiological conditions (*Wiencke et al., 2020*). Based on the calculated rate constant for OCT2 in vivo and the known water space of average cells, cytoplasmic concentrations of DA at equilibrium are thought to be ~10-fold higher compared to the extracellular concentrations (*Gründemann et al., 1998*). For instance, Grundemann et al. have shown that addition of 100 nM DA in the extracellular environment of OCT2-expressing cells results in the accumulation of 4 pmol/mg in cells after 10 min. If we consider the average weight of a cell to be around 1 ng, and the average volume of a cell to be around 4 pL, this calculation will give us close to 1 µM DA accumulation in the cytoplasm, which is 10-fold higher than the added extracellular concentration (*Gründemann et al., 1998*). We found that the requirement for OCT2 in activating Golgi-localized D1DRs is seen

even at low concentrations of exogenously added DA (10 nM) (*Figure 1—figure supplement 3*). Notably, knockdown or inhibition of OCT2 abrogated Golgi-localized D1DR signaling (*Figures 2 and 3*), highlighting the specificity of OCT2 in this signaling regulation.

The present results expand the concept of GPCR-compartmentalized signaling and open additional interesting questions for further studies regarding mechanisms that regulate subcellular activity of other monoamine receptors such as 5-HT (serotonin) and histamine receptors by other monoamine transporters (*Lin et al., 2011*; *Torres et al., 2003*). Establishing GPCR signaling from subcellular compartments is the first step in unraveling the physiological consequences of compartmentalized signaling for each GPCR family member.

# Materials and methods

**Key resources table**

| Reagent type (species) or resource | Designation | Source or reference | Identifiers | Additional information |
|---|---|---|---|---|
| Cell line (*Homo sapiens*) | HEK293 | ATCC | CRL-1573.3 | Mycoplasma tested negative |
| Cell line (*H. sapiens*) | HEK293T | ATCC | ACS-4500 | Mycoplasma tested negative |
| Cell line (*H. sapiens*) | HeLa | ATCC | CRM-CCL-2 | Mycoplasma tested negative |
| Antibody | Anti-dopamine receptor D1(rabbit polyclonal) | Abcam | ab216644 | IF (1:100) |
| Antibody | Anti-GM130 (mouse monoclonal) | BD Biosciences | 610822 | IF (1:1000) |
| Antibody | Anti-SLC22A2 (rabbit polyclonal) | Abcam | ab170871 | WB (1:1000) |
| Antibody | Anti-SLC22A2 (rabbit polyclonal) | ABclonal | A14061 | IF (1:100) |
| Antibody | Anti-GAPDH (mouse monoclonal) | Proteintech | 60004-1 | WB (1:10,000) |
| Antibody | Anti-HRP-conjugated IgG (rabbit polyclonal) | GE Healthcare | P132460 | WB (1:10000) |
| Antibody | Anti-mouse IgG (donkey polyclonal) | Thermo Fisher | A32766 | IF (1:10,000) |
| Antibody | Anti-rabbit IgG (donkey polyclonal) | Thermo Fisher | A32794 | IF (1:10,000) |
| Antibody | Anti-dopamine receptor D1 (rabbit polyclonal) | Proteintech | 17934-1-AP | IF (1:100) |
| Peptide, recombinant protein | D1DR blocking peptide | Proteintech | Ag12366 | IF (1:25) |
| Chemical compound, drug | Snap-Cell 647 | NEB | S9102S | |
| Chemical compound, drug | Dopamine hydrochloride | Sigma | 200-527-8 | |
| Chemical compound, drug | SKF81297 hydrobromide | Tocris | 1447 | |
| Chemical compound, drug | NPEC-caged-dopamine | Tocris | 3992 | |
| Chemical compound, drug | A/C heterodimerizer | Takara | 635056 | |
| Chemical compound, drug | Dyngo | Abcam | ab120689 | |
| Chemical compound, drug | Forskolin | Sigma | F6886-10MG | |
| Chemical compound, drug | Imipramine | Sigma | 113-52-0 | |
| Chemical compound, drug | Corticosterone | Sigma | 200-019-6 | |
| Software, algorithm | Prism | GraphPad | | |
| Software, algorithm | ImageJ | Imagej.net/contributors | | |

*Continued on next page*

Continued

| Reagent type (species) or resource | Designation | Source or reference | Identifiers | Additional information |
|---|---|---|---|---|
| Software, algorithm | MATLAB R2014b | MathWorks | DOI:10.5281/zenodo.5146169 | *Bakr et al., 2021* |
| Recombinant DNA reagent | pGloSensor-20F | Promega | E1171 | |
| Recombinant DNA reagent | Signal sequence Snap-D1DR | This study | | pCDNA3 backbone; snap vector, see Materials and methods |
| Recombinant DNA reagent | pVenus-FRB-Nb6B9 | This study | | pVenus-C1 vector, see Materials and methods |
| Recombinant DNA reagent | FKBP-GalT-mApple | This study | | pm-Apple-M1 vector, see Materials and methods |
| Recombinant DNA reagent | Lyn-2xFKBP-CFP | Addgene | 20149 | |
| Recombinant DNA reagent | pCAG-Snap-D1DR | This study | | pCAG vector, see Materials and methods |
| Recombinant DNA reagent | pCAG-Nb6B9-GFP pCAG-Nb37-GFP | This study | | pCAG vector, see Materials and methods |
| Sequence-based reagent | SLC22A2 shRNAs | OriGene | TL517269 | pGFP-C-shLenti |
| Sequence-based reagent | Scamble shRNAs | OriGene | TR30021 | pGFP-C-shLenti |
| Biological sample (*Mus musculus*, male and female) | CD1 | Charles Rivers | Crl:CD1(ICR) | Isolated medium spiny neurons from neonatal mouse striatal, see Material and methods |

## Cell culture, cDNA constructs, and transfection

HeLa and HEK293 cells (purchased from ATCC as authenticated lines CCL-2, CRL-1573 and CRL 1446, respectively) were grown in Dulbecco's minimal essential medium supplemented with 10% fetal bovine serum (FBS) without antibiotics. Cell cultures were free of mycoplasma contamination. Signal Sequence-Snap-tagged D1DR was created by amplifying D1DR from Flag-D1DR using 5′-GCCT GGGCTGGGTCTTGGATCCGATGACGCCATGGACG -3′; 5′-ATAGGGCCCTCTAGAGCCTCAGGT TGGGTGCTG-3′ primers, and inserted into the Snap vector using BamHI and XbaI. pVenus-FRB-Nb6B9 was created by amplifying Nb6B9 and FRB from Nb6B9-GFP (*Irannejad et al., 2017*), and pC$_4$-R$_H$E plasmid (ARIAD Pharmaceuticals), using 5′-TGGTGGACAGGTGCAGCT-3′; 5′- GGATCCTC ATGAGGAGACGGTGACCTGGGT-3′ and 5′-GCTTCGAATTCAATCCTCTGGCAT-3′; 5′-TGCACCTGTC CACCAGCACTA-3 primers, respectively, such that it contained the linker sequence GATAGTGCTGGT AGTGCTGGTGGAC, and inserted into the pVenus-C1 vector using EcoRI and BamH1. FKBP-GalT-mApple was created by amplifying FKBP and GalT from KDELr-FKBP and GalT-mCherry plasmids (a generous gift from Dr.Farese lab), using 5′-CATGCTAGCGCCGCCACCATGGGAGTGCAGGTGGAA ACCAT-3′, 5′-GAGCTCGAGACCAGCACTACCAGCACTATCCTCCAGCTTCAGCAGCTCCACG3′ and 5′- GCTCAAAGCTTGCCGCCACCGGAAGGCTTCGGGAGCCG-3′, 5′-ACCGGATCCTTAGGCCCCTC CGGTCCGGAGCTCCCCG-3′ primers, respectively, and inserted into the pmApple-N1 vector using NheI, XhoI for FKBP and HindIII and BamHI for GalT (*Irannejad et al., 2017*). Transfections were performed using Lipofectamine 2000 (Invitrogen) according to the manufacturer's instructions. Snap-tagged human D1DR constructs were labeled with Snap-cell 647 SiR (New England Biolabs, S9102S) as described previously (*Lukinavičius et al., 2013*).

## Isolation of murine striatal neurons

Primary striatal neurons were prepared from P1-P2 CD1 pups. In brief, striatum tissues isolated from brains in cold HBSS (w/o Mg$^{2+}$, Ca$^{2+}$, and phenol-red) buffer with 10 mM HEPES were treated by HBSS with 0.25% Trypsin and 10 mM HEPES buffer at 37°C for 15 min. The digested striatum tissues were rinsed by neural plating media (DMEM with 10% FBS, 30 mM HEPES, and PS) twice, and then dissociated by trituration using fire-polished Pasteur pipet in neural plating media. Suspended cells that pass through a 40 µm strainer were collected by centrifuging at 350 × *g* for 5 min. Cells were plated at 10$^3$ cells per mm$^2$ on the 100 µg/mL poly-D-lysine (Sigma)-coated imaging dishes or coverslips in neural plating media. After 16–24 hr, the culture media were replaced by neural differentiation media

(Neural basal media with 10 mM GlutaMAX, B27, and PS). The 50% media were replaced by fresh neural differentiation media every 3–4 days.

## Live-cell confocal imaging

Live-cell imaging was carried out using Nikon spinning disk confocal microscope with a ×60, 1.4 numerical aperture, oil objective and a $CO_2$ and 37°C temperature-controlled incubator. A 488, 568 nm and 640 Voltran was used as light sources for imaging GFP, mRFP/mApple, and Snap-647 signals, respectively. Cells expressing both Snap-tagged receptor (2 µg) and the indicated nanobody–GFP (200 ng) were plated onto glass coverslips. Receptors were surface labeled by addition of Snap-Cell 647 SiR (1:1000, New England Biolabs) to the media for 20 min, as described previously. Live-cell images where endocytosis was inhibited were carried out by incubating the cells in 30 µM Dyngo 4a (ab120689) at 37°C for 30 min before indicated agonist was added. HEK293 PKA-Cat-GFP knock-in cells were a generous gift from the Huang Lab. Indicated agonists (dopamine hydrochloride [Sigma], SKF81297 hydrobromide [Tocris]) were added and cells were imaged every 20 s for 20 min in DMEM without phenol red supplemented with 30 mM HEPES, pH 7.4. NPEC-caged-dopamine (Tocris) was incubated for 10 min before cells were stimulated with 3.2 µW/cm² blue light. Time-lapse images were acquired with a CMOS camera (Photometrics) driven by Nikon Imaging Software (NS Elements).

## Fixed-cell confocal imaging

Cells were permeabilized with saponin to reduce the cytoplasmic background, as described previously (*Lobert and Stenmark, 2012*). Briefly, HeLa cells were permeabilized with 0.05% saponin (Sigma) in PEM buffer (80 mM K-Pipes, pH 6.8, 5 mM EGTA, 1 mM $MgCl_2$) for 5 min on ice. Cells were then fixed with 3% paraformaldehyde in PBS for 10 min and then quenched with 50 mM $NH_4Cl$ in PBS for 15 min. Primary antibodies D1DR antibody (ab216644) (1:100), D1DR (Proteintech 17934-1AP) (1:100) with or without D1DR blocking peptide (Proteintech Ag12366) (1:25), GM130 (BD Biosciences 610822) (1:1000), and SLC22A2/OCT2 antibody (ab170871) or SLC22A2/OCT2 (ABClonal-A14061) (1:100), were diluted in PBS supplemented with 0.05% saponin. Striatal neurons at DIV5 were fixed by 3.7% formaldehyde in PEM buffer for 15 min and then permeabilized by 0.3% Triton in PBS for 5 min at room temperature. D1DR and GM130 antibodies were diluted in TBS with 5% donkey serum and 0.1% Triton X-100. Confocal images were taken using Nikon spinning disk confocal microscope with a 60 × 1.4 numerical aperture, oil objective.

## Image analysis and statistical analysis

Images were saved as 16-bit TIFF files. Quantitative image analysis was carried out on unprocessed images using ImageJ software (http://rsb.info.nih.gov/ij). For measuring kinetics of Nb6B9–GFP and miniGs recruitment at the Golgi membrane over time in confocal images and kinetics of PKA-Cat GFP dissociation from the Golgi, analyses were performed on unprocessed TIFF images using a previously published scripts written in MATLAB, available through open access on zenoob (*Bakr et al., 2021*; *Jullié et al., 2020*). Briefly, the Golgi region was selected and a mask of labeled receptor (using Snap label) or Golgi marker were generated by thresholding the receptor or the Golgi marker signal within the selected region. The average fluorescence intensity of Nb6B9 or miniG were measured within the masked region and outside of the masked region, before and after addition of agonists. Values were normalized by calculating the percent relative to the maximum value, then baseline corrected using Prism 6.0 software so that the first value of each condition was set to 0. The same MATLAB script was used to analyze the dose–response kinetics of both Nb6B9-GFP and miniGs-mApple recruitment to the Golgi membrane in response to increasing concentrations of agonists. In this case, to better quantify the increase in fluorescence at the Golgi after addition of agonist, values were normalized to the baseline following each addition of agonist. This was done in Microsoft Excel, and each baseline value was set to 1 to measure the fold change in fluorescence. Colocalization analysis at the Golgi was estimated by calculating the Pearson's coefficient between the indicated image channels with the Golgi marker channel using the colocalization plug-in for ImageJ (Coloc2). p-Values are from two-tailed unpaired Student's *t*-tests calculated using Prism 6.0 software (GraphPad Software).

## Luminescence-based cAMP assay

HEK293 cells stably expressing D1DR were transfected with a plasmid encoding a cyclic-permuted luciferase reporter construct (pGloSensor-20F, Promega) and luminescence values were measured, as described previously (*Irannejad et al., 2013*). Briefly, cells were plated in 96-well dishes (~100,000 cells per well) in 500 μL DMEM without phenol red/no serum and equilibrated to 37°C in the SpectraMax plate reader and luminescence was measured every 1.5 min. Software was used to calculate integrated luminescence intensity and background subtraction. In rapamycin heterodimerization experiments, cells were pre-incubated with 1 μM A/C heterodimerizer, a rapamycin analog (Takara 635056) for 15 min. 5 μM forskolin was used as a reference value in each multi-well plate and for each experimental condition. The average luminescence value (measured across duplicate wells) was normalized to the maximum luminescence value measured in the presence of 5 μM forskolin. For rapamycin-treated cells, the average luminescence value was normalized to the maximum luminescence value measured in the presence of 5 μM forskolin and 1 μM rapamycin.

## Western blotting

Cells from HEK293, HEK293T, and HeLa were lysed in extraction buffer (0.2% Triton X-100, 50 mM NaCl, 5 mM EDTA, 50 mM Tris at pH 7.4 and cOmplete EDTA-free Protease Inhibitor Cocktail [Roche]). Kidney and neural tissues from B6 adult mice were disrupted in RIPA buffer (50 mM Tris at pH 7.4, 150 mM NaCl, 1 mM EDTA, 1% Triton X-100, 1% sodium deoxycholate 0.1% SDS, and cOmplete EDTA-free Protease Inhibitor Cocktail). After agitation at 4°C for 30 min, supernatants of samples were collected after centrifuging at 15,000 × rpm for 10 min at 4°C. Supernatants were mixed with SDS sample buffer for the protein denaturation. The proteins were resolved by SDS-PAGE and transferred to PVDF membrane and blotted for anti-SLC22A2 (ab170871, 1:1000) or GAPDH (1:10,000) antibodies to detect OCT2 and GAPDH expression by horseradish-peroxidase-conjugated rabbit IgG, sheep anti-mouse and rabbit IgG (1:10,000 Amersham Biosciences), and SuperSignal extended duration detection reagent (Pierce).

## Acknowledgements

We thank D Jullie, K Silm, AB Lobingier, A Manglik, E Hernandez, and D Larsen for assistance, advice, and valuable discussion. These studies were supported by the National Institute on General Medicine (GM133521) to RI and The American Heart Association (908933) to NP.

## Additional information

### Funding

| Funder | Grant reference number | Author |
| --- | --- | --- |
| NIH Office of the Director | GM133521 | Roshanak Irannejad |
| The American Heart Association | 908933 | Natasha M Puri |

The funders had no role in study design, data collection and interpretation, or the decision to submit the work for publication.

### Author contributions

Natasha M Puri, Conceptualization, Data curation, Formal analysis, Investigation, Methodology, Validation, Visualization, Writing - original draft, Writing - review and editing; Giovanna R Romano, Formal analysis, Investigation, Software, Validation, Visualization; Ting-Yu Lin, Formal analysis, Methodology, Validation, Visualization, Writing - review and editing; Quynh N Mai, Data curation, Formal analysis, Methodology, Validation, Visualization, Writing - review and editing; Roshanak Irannejad, Conceptualization, Formal analysis, Funding acquisition, Investigation, Methodology, Software, Supervision, Validation, Writing - original draft, Writing - review and editing

### Author ORCIDs

Natasha M Puri http://orcid.org/0000-0002-4290-4361
Quynh N Mai http://orcid.org/0000-0002-6199-2096

Roshanak Irannejad ⓘ http://orcid.org/0000-0001-8702-2285

### Ethics

Animal tissues from mice were isolated as recommended and approved by the institutional animal care and use committee (IACUC protocol #AN184251).

### Decision letter and Author response

Decision letter https://doi.org/10.7554/eLife.75468.sa1
Author response https://doi.org/10.7554/eLife.75468.sa2

---

## Additional files

### Supplementary files

• Transparent reporting form

• Source data 1. Source data of all figures and figure supplements are combined in one file, separated and labeled accordingly in each tab.

### Data availability

Source Data has been provided for Figures 1c, 1d, 2b, 2c, 2f, 2g, 3c, 4d, 4f, 5c and 6c as well as Figures 1-Figure Supplement 1b-d, Figure 1-Figure Supplement 3b, Figure 1-Figure Supplement 4c, Figure 2-Figure Supplement 1a-b, Figure 3-Figure Supplement 1a. We have also included information for primers, shRNAs and plasmid maps. Antibodies, cells and reagents are also provided in the source data.

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
