## [Editor Report]

This study uses a tour de force of biosensor constructs providing evidence that dopamine transport by OCT2 across the plasma membrane and also (presumably) into the Golgi activates GPCR signaling at the Golgi, leading to cAMP production and PKA activation. Thus, intracellularly compartmentalized signaling underlies aspects of dopamine D1 receptor signaling. The work will be of interest to scientists working on the biology of dopamine signaling.

---

## [Decision Letter]

**Decision letter after peer review:**

Thank you for submitting your article "The OCT2 Transporter Regulates Dopamine D1 Receptor Signaling at the Golgi Apparatus" for consideration by *eLife*. Your article has been reviewed by 3 peer reviewers, one of whom is a member of our Board of Reviewing Editors, and the evaluation has been overseen by Vivek Malhotra as the Senior Editor. The following individual involved in review of your submission has agreed to reveal their identity: Paul Gasser (Reviewer #2).

Essential revisions:

This manuscript will be of interest to cell biologists studying signal transduction, and to physiologists and neuroscientists studying mechanisms of dopamine action. It describes a novel signaling mechanism that may contribute to the actions of dopamine. The key conclusions of the manuscript are supported by the data. However, the reviewers identified issues (see the full reviews below) that should be addressed. Most of the issues are relatively straightforward. The essential points are:

1. The reviews identify the need for a clear demonstration that endogenous OCT2 and D1DR are Golgi localized.

2. Several of the review issues concern specificity of reagents used including corticosterone, imipramine, dopamine concentration (micro vs nanomolar), Nb6B9 and the anti-D1DR antibody.

3. Showing that endogenous D1DR can recruit Nb37 or release PKA in an OCT2-dependent manner would be a step towards establishing physiological relevance. Why was D1DR expressed in MSNs in these experiments? Were endogenous D1DR not able to be activated?

4. A critical aspect is determining the physiological relevance of this pathway. For β adrenergic receptor the authors state "Golgi-b1ARs only, but not the plasma membrane pool, lead to PLCe activation at the perinuclear/Golgi membrane, which mediates hypertrophic responses in cardiomyocytes (40, 41)". It is recommended that the current manuscript is clear about the state of knowledge of the role for Golgi signaling by D1DR. They might also expand on their speculation that Golgi D1DR so that readers understand why Golgi D1DR might mediate sustained cAMP/PKA signaling and what the purpose of this would be?*Reviewer #1 (Recommendations for the authors):*

My review of an initial submission highlighted the need for a physiologically relevant cell type, the issue of showing that OCT2 is not only sufficient but also required for activation of Golgi-localized receptor, and a concern about how cytoplasmic dopamine gains access to the Golgi lumen. While the latter concern somewhat remains, this version satisfactorily addresses these issues and I believe the study will be of interest to a large audience.

I'm mostly satisfied with the improvements of the present version. The authors state "OCT transporters are localized on the plasma membrane and intracellular compartments including nuclear envelope; thus they can transport catecholamines across the plasma membrane and across internal membrane compartments (40, 49)." Evidence that OCT2 is partly Golgi-localized seems to be currently lacking. The experiment seems straightforward and would go a long way to providing an explanation for how cytoplasmic dopamine ends up in the Golgi lumen.*Reviewer #2 (Recommendations for the authors):*

The phrase, "OCT2 transporter", which is used in the title and throughout the manuscript, is redundant. In the title, it should be replaced with "organic cation transporter 2". In the rest of the manuscript, it could be replaced by "OCT2".

There are some statements in the manuscript that oversimplify aspects of dopamine function.

– Abstract "Dopamine is the main catecholamine in the brain and kidney…" This statement is not accurate, as epinephrine and norepinephrine are abundant and critically important catecholamines. Dopamine is important, but is not the "main" catecholamine.

– Abstract: "… where it controls a number of physiological functions…" This oversimplifies the role of dopamine. It is involved in the regulation of these functions, but does not "control" them.

– Introduction: "Several pathological conditions such as…. are due to dysregulation of the neuronal dopaminergic signaling pathway." This statement significantly oversimplifies the current understanding of the pathophysiology of all of these disorders. They involve dysregulation of dopaminergic signaling but are not "due to" dysregulation of dopamine signaling.

– "moving defects" of Parkinson's is better described as motor deficits.

Page 6: in the description of the effects of Nb6B9 on SKF-induced cAMP responses, the authors state that treatment with rapalog in cells expressing Golgi-targeted FKBP and FRB-Nb6B9 and subsequent stimulation with SKF81297 "resulted in only activation of the plasma membrane D1R pool (Figure 4e)." The data in Figure 4e do not demonstrate that SKF treatment under these conditions only activated the plasma membrane pool. They merely show that the cAMP response to SKF was smaller under these conditions.

Page 6: Regarding the presence of an OCT2-immunoreactive band in western blots from HEK293T cells: The fact that HEK cells do express some level of OCT2 does not invalidate the authors' conclusions about its role in mediating Golgi D1 activation. This could be an opportunity to discuss the importance of knowing the subcellular localization of transporters. Given the likelihood that the ligand-binding site of the Golgi D1r is in the lumen of the Golgi, dopamine would have to be transported across both plasma and Golgi membranes to activate the receptors.

In instances when a Golgi marker is not used (all of the live imaging studies in cultured striatal neurons), the authors use SNAP-D1 to identify Golgi localized receptors. This requires the assumption that the perinuclear pool of D1 fluorescence is indeed at the Golgi. It is fine to make this assumption, but the authors should make the link to the immunostaining they did to justify the assumption.

The description of primary cell culture methods is entitled, "Isolation of murine striatal and hippocampal neurons". There are no studies involving hippocampal neurons in the manuscript.

*Reviewer #3 (Recommendations for the authors):*

Showing that endogenous D1DR can recruit Nb37 or release PKA in an OCT2-dependent manner in the would be a step towards establishing physiological relevance. Why was D1DR expressed in MSNs in these experiments? Were endogenous D1DR not able to be activated?

Arrowhead on Figure 1 is described as denoting active D1DR at Golgi membrane, but the spot pointed to by the arrowhead does not express the Golgi marker. What is this membrane?

In the document given for review, the images seem blurry. Were they image processed, e.g. smoothened?

Are the normalized fluorescence traces for HEK293 in Figure 1 and 2 the same data shown twice? Ideally these should be controls paired to the experimental conditions.

---

## [Author Response]

Essential revisions:This manuscript will be of interest to cell biologists studying signal transduction, and to physiologists and neuroscientists studying mechanisms of dopamine action. It describes a novel signaling mechanism that may contribute to the actions of dopamine. The key conclusions of the manuscript are supported by the data. However, the reviewers identified issues (see the full reviews below) that should be addressed. Most of the issues are relatively straightforward. The essential points are:1. The reviews identify the need for a clear demonstration that endogenous OCT2 and D1DR are Golgi localized.

We have now included additional data demonstrating endogenous localization of OCT2 at the Golgi in addition to the plasma membrane in both HeLa and MSNs (Figure 2—figure supplement 1d and Figure 3—figure supplement 1d). We have confirmed the specificity of the antibody using shRNA against OCT2 (Figure 2e and Figure 2—figure supplement 1d).

2. Several of the review issues concern specificity of reagents used including corticosterone, imipramine, dopamine concentration (micro vs nanomolar), Nb6B9 and the anti-D1DR antibody.

Imipramine: We agree that higher concentration of imipramine can have non-specific effects. Therefore, we now use a lower concentration of imipramine (10µM) and show that D1DR activation at the Golgi is still abrogated (Figure 2b). Please note that we now also use shRNA to directly inhibit OCT2.

Corticosterone-mediated repression of OCT3: We previously reported that HeLa cells also express OCT3 and blocking its activity with 10µM corticosterone inhibits epinephrine-mediated activation of the Golgi-localized β1AR (PMID: 28553949). As Reviewer #2 indicated, corticosterone has been reported to also inhibit OCT2-mediated uptake of DA in a stable transfection system with a Ki Value of 500nM (PMID: 9812985). However, in our hands, we did not observe inhibition of D1DR signaling at the Golgi when cells were treated with 10µM corticosterone (Figure 2b). To bolster the validity of our conclusions and the specificity of OCT2 in transporting dopamine in HeLa cells, we have included additional data using two different shRNAs against OCT2 to show that genetic knock-down of OCT2 in HeLa cells blocks D1DR activation at the Golgi. By contrast, control (scrambled) shRNA had no effect on D1DR activation at the Golgi, suggesting specificity of OCT2 shRNA transfection in HeLa cells (Figure 2—figure supplement 1b). Importantly, SKF81297, a membrane permeant agonist that diffuses across the membrane and does not require OCT2, can still reach the Golgi membranes and activate D1DR at the Golgi even when OCT2 is genetically knocked down (Figure 2 d-g).

Dopamine and Nb6B9: To better quantify D1DR activation at the plasma membrane and the Golgi in response to various doses of dopamine, we have included new data. Increased concentrations of dopamine were added to the same cells over time and Nb6B9 or miniG_s_ recruitments to the plasma membrane and the Golgi were quantified. Nb6B9 recruitment quantifications are shown in Figure 1c and Figure 1—figure supplement 1b for the Golgi and the plasma membrane, respectively. MiniG_s_ recruitment quantifications are shown in Figure 1—figure supplement 3b. We were able to detect plasma membrane activation of D1DR starting at 10nM dopamine concentration using both biosensors. Subtle D1DR activation at the Golgi were detected at 10nM and 100nM dopamine by miniG_s_ and Nb6B9, respectively. Although we do not have exact measurements for Nb6B9 versus miniG_s_ binding affinities to activated D1DR, these observations suggest that miniG_s_ is more sensitive in detecting activated D1DR. It is important to clarify that our ability to precisely measure D1DR activation by low concentrations of agonists at the Golgi is limited due to higher cytoplasmic background of biosensors that mask their earlier recruitment to the Golgi. Therefore, to better quantify the increase in fluorescence intensity at the Golgi after addition of agonist, values were normalized to the baseline following each addition of agonist. Each baseline value was set to 1 to measure the fold change in fluorescence. These calculations were done using Microsoft Excel. At 10 nM agonist concentration we first observe the plasma membrane recruitment of the biosensors. As a result, when calculating fluorescence intensity of the biosensor recruitment to the Golgi, we initially observe a decrease in the cytoplasmic fluorescence due to biosensor recruitment to the plasma membrane, followed by an increase in the Golgi recruitments (Figure 1c, Figure 1—figure supplement 1c and Figure 1—figure supplement 3b).

Detection of β1AR activation at the plasma membrane and the Golgi by Nb6B9 occurred at comparable epinephrine concentrations (Figure 1—figure supplement 1c and d). Importantly, Nb6B9 was unable to detect activation of Gi-coupled GPCRs such as δ opioid receptors (Figure 1—figure supplement 1e), indicating its specificity of binding to catecholamine receptors where Nb6B9 binding sites are conserved (Figure 1—figure supplement 1a).

D1DR antibody: To test the specificity of D1DR antibody used to detect endogenous D1DR localization in MSNs, we used a commercially available D1DR antibody that has been validated by immunostaining. Using this new antibody, we were able to detect endogenous D1DR on both the plasma membrane and the Golgi membranes in MSNs (Figure 3—figure supplement 1c). Importantly, D1DR immunostaining was largely diminished when MSNs were immuno-stained in the presence of D1DR blocking peptide (Figure 3—figure supplement 1c). Of course, we used the same laser power and exposure time to image our samples by a spinning disk microscope.

3. Showing that endogenous D1DR can recruit Nb37 or release PKA in an OCT2-dependent manner would be a step towards establishing physiological relevance. Why was D1DR expressed in MSNs in these experiments? Were endogenous D1DR not able to be activated?

All of the previously reported nanobody-based biosensors and miniG proteins (PMID: 23515162, PMID: 28553949, PMID: 29754753, PMID: 29523687, PMID: 31263273) have relied on conditions where target receptors are over-expressed. This is because endogenous GPCRs are expressed at very low levels and the cytoplasmic levels of nanobody or miniG-based biosensors present high levels of background. This high background prevents easy visualization of biosensor recruitment to any membranes, including the plasma membrane. Thus, in order to achieve a higher signal-to-noise ratio and to increase the level of detection, receptor expression has been increased. While the endogenous receptor activity can potentially be monitored by employing super resolution microscopy techniques such as TIRF microscopy to track single-particles of photo-switchable nanobodies (PMID: 29045395, PMID: 33214152), this type of TIRF microscopy is only suitable for cell surface receptors, and not applicable to receptors located at internal membrane locations such as the Golgi.

It was commented that Nb37 shows much weaker recruitment to D1DR/G_s_ receptors compared to NB6B9. This is expected and consistent with measured binding affinities of the two biosensors: we previously found that the binding affinity of Nb37 to agonist bound β2AR/G_s_ complex is ~800nM. By comparison, binding affinities of Nb80 and Nb6B9 (receptor nanobodies) for agonist bound β2AR have been reported at ~10nM (PMID: 23515162, PMID: 24056936). Although we do not have the exact affinity measurements of Nb6B9 and Nb37 for D1DR, our dose dependent recruitment experiments suggest comparable binding affinities of these biosensors to the D1DR/G_s_ complex as those reported for the β2AR/G_s_ complex (Figure 1c and Figure 1—figure supplement 1b-d).

We now provide new data demonstrating that D1DR can form a functional complex with Gs protein at the Golgi in MSNs. To improve visualization of the weak Nb37 signal, we fixed and permeabilized MSNs and immune-stained them with GFP antibodies. Our new data in Figure 3c shows Nb37-GFP recruitment to activated D1DR in MSNs after 5 min 10µM DA stimulation, providing evidence for functional G protein coupling of D1DR in physiologically relevant cell type.

4. A critical aspect is determining the physiological relevance of this pathway. For β adrenergic receptor the authors state "Golgi-b1ARs only, but not the plasma membrane pool, lead to PLCe activation at the perinuclear/Golgi membrane, which mediates hypertrophic responses in cardiomyocytes (40, 41)". It is recommended that the current manuscript is clear about the state of knowledge of the role for Golgi signaling by D1DR. They might also expand on their speculation that Golgi D1DR so that readers understand why Golgi D1DR might mediate sustained cAMP/PKA signaling and what the purpose of this would be?

Thank you for this suggestion. We have now revised the discussion and expanded our speculation on the role of sustained cAMP/PKA signaling in regulating cellular and physiological responses. Below is the summary of what is revised in the paper:

D1DR-mediated cAMP signaling regulates major brain functions and persistence DA stimulation and sustained receptor activation causes long-term changes in gene expression of neuronal plasticity-related genes, dendritic remodeling and locomotor sensitization. Many drugs of abuse, increase the release of DA and elevate the firing rate of midbrain dopaminergic neurons in the striatum, particularly MSNs. The recognition of subcellular D1DR activity could have major implications for physiological processes regulated by DA. As we have discussed in the paper, the importance of cAMP generation by Golgi-localized GPCR has been demonstrated for β1AR, where Golgi-β1ARs only, but not the plasma membrane pool, lead to PLC_Ɛ_ activation at the perinuclear/Golgi membrane, which mediates hypertrophic responses in cardiomyocytes (PMID: 31433293). Whether distinct signaling pathways are regulated by plasma membrane or Golgi-localized D1DR is not yet clear.

Our data also suggest that D1DR activation at the Golgi is required for sustained cAMP/PKA response (Figure 5). Whether Golgi-localized D1DR has a distinct role in regulating sustained cAMP/PKA response is not clear but strongly suggested by our findings. Given the higher expression of OCT2 in striatum and the reported evidence for sustained cAMP/PKA signaling in MSNs compared to cortical neurons (PMID: 23551948), we speculate that activation of the Golgi-pool of D1DRs may be a determinant of which cell types and tissues exhibit sustained cAMP/PKA signaling. However, the precise experimental evidence for this hypothesis is beyond the scope of this paper and we hope to further explore that in the future. This speculation and our current understanding of D1DR subcellular signaling is now included in the discussion.

Reviewer #1 (Recommendations for the authors):My review of an initial submission highlighted the need for a physiologically relevant cell type, the issue of showing that OCT2 is not only sufficient but also required for activation of Golgi-localized receptor, and a concern about how cytoplasmic dopamine gains access to the Golgi lumen. While the latter concern somewhat remains, this version satisfactorily addresses these issues and I believe the study will be of interest to a large audience.I'm mostly satisfied with the improvements of the present version. The authors state "OCT transporters are localized on the plasma membrane and intracellular compartments including nuclear envelope; thus they can transport catecholamines across the plasma membrane and across internal membrane compartments (40, 49)." Evidence that OCT2 is partly Golgi-localized seems to be currently lacking. The experiment seems straightforward and would go a long way to providing an explanation for how cytoplasmic dopamine ends up in the Golgi lumen.

We are grateful for the positive comments and that the reviewer deems this version of the manuscript as improved. To address the remaining concern of the reviewer, we have now added data (Figure 2—figure supplement 1d and 7d) to show endogenous OCT2 localization at the Golgi in HeLa cells and MSNs. Importantly, we have confirmed the specificity of our OCT2 antibody by showing that immunostaining is abrogated in cells expressing OCT2 shRNAs (Figure 2—figure supplement 1d).

Reviewer #2 (Recommendations for the authors):The phrase, "OCT2 transporter", which is used in the title and throughout the manuscript, is redundant. In the title, it should be replaced with "organic cation transporter 2". In the rest of the manuscript, it could be replaced by "OCT2".

Thank you for pointing this out. This has been corrected.

There are some statements in the manuscript that oversimplify aspects of dopamine function.– Abstract "Dopamine is the main catecholamine in the brain and kidney…" This statement is not accurate, as epinephrine and norepinephrine are abundant and critically important catecholamines. Dopamine is important, but is not the "main" catecholamine.

This sentence has been corrected in the abstract.

– Abstract: "… where it controls a number of physiological functions…" This oversimplifies the role of dopamine. It is involved in the regulation of these functions, but does not "control" them.

This sentence has been edited.

– Introduction: "Several pathological conditions such as…. are due to dysregulation of the neuronal dopaminergic signaling pathway." This statement significantly oversimplifies the current understanding of the pathophysiology of all of these disorders. They involve dysregulation of dopaminergic signaling but are not "due to" dysregulation of dopamine signaling.

This sentence has been revised.

– "moving defects" of Parkinson's is better described as motor deficits.

This has been edited.

Page 6: in the description of the effects of Nb6B9 on SKF-induced cAMP responses, the authors state that treatment with rapalog in cells expressing Golgi-targeted FKBP and FRB-Nb6B9 and subsequent stimulation with SKF81297 "resulted in only activation of the plasma membrane D1R pool (Figure 4e)." The data in Figure 4e do not demonstrate that SKF treatment under these conditions only activated the plasma membrane pool. They merely show that the cAMP response to SKF was smaller under these conditions.

Thank you for pointing this out. We have now revised this sentence to reflect the correct interpretation of the data.

Page 6: Regarding the presence of an OCT2-immunoreactive band in western blots from HEK293T cells: The fact that HEK cells do express some level of OCT2 does not invalidate the authors' conclusions about its role in mediating Golgi D1 activation. This could be an opportunity to discuss the importance of knowing the subcellular localization of transporters. Given the likelihood that the ligand-binding site of the Golgi D1r is in the lumen of the Golgi, dopamine would have to be transported across both plasma and Golgi membranes to activate the receptors.

As mentioned above, we have now added new data (Figure 2—figure supplement 1d) showing immunostaining of endogenous OCT2 in Hela cells at the PM and the Golgi. We failed to detect OCT2 expression in HEK293 cells. This is also functionally shown in Figure 1b, where even a high concentration of Dopamine (10µM) fails to activate D1DR at the Golgi. This is why we believe that the low expression of OCT2 in HEK293 cells is insufficient for DA delivery to the Golgi-D1DR.

In instances when a Golgi marker is not used (all of the live imaging studies in cultured striatal neurons), the authors use SNAP-D1 to identify Golgi localized receptors. This requires the assumption that the perinuclear pool of D1 fluorescence is indeed at the Golgi. It is fine to make this assumption, but the authors should make the link to the immunostaining they did to justify the assumption.

Thank you for the suggestion. We have shown in multiple examples that our Snap-tagged D1DR detects Golgi-localized D1DR in live imaging experiments (Figure 1, 2 and 4 and Figure 1-Video 1-4). This is specifically shown in MSNs by immunostaining shown in Figure 3—figure supplement 1b top panel. We have now clarified this and have referenced the figure separately when describing the data in the main text.

The description of primary cell culture methods is entitled, "Isolation of murine striatal and hippocampal neurons". There are no studies involving hippocampal neurons in the manuscript.

This has been corrected.

Reviewer #3 (Recommendations for the authors):Showing that endogenous D1DR can recruit Nb37 or release PKA in an OCT2-dependent manner in the would be a step towards establishing physiological relevance. Why was D1DR expressed in MSNs in these experiments? Were endogenous D1DR not able to be activated?

The point regarding the need to overexpress D1DR in MSNs has been described above. We now provide new data demonstrating that D1DR can form a functional complex with G protein at the Golgi in MSNs, using Nb37-GFP. To improve visualization of the weak Nb37 signal, we fixed and permeabilized MSNs and immuno-stained them with GFP antibodies. Our new data in Figure 3c shows Nb37-GFP recruitment to activated D1DR in MSNs after 5 min 10µM DA stimulation, providing evidence for functional G protein coupling of D1DR in physiologically relevant cell type.

Arrowhead on Figure 1 is described as denoting active D1DR at Golgi membrane, but the spot pointed to by the arrowhead does not express the Golgi marker. What is this membrane?

The arrowhead placement has now been corrected to better show biosensor recruitment to the Golgi. Nb6B9 and miniGs are recruited to the Golgi membrane as we have shown repeatedly in our paper using a Golgi marker (Figure 1b, Figure 2a, Figure 1—figure supplement 2a, Figure 1—figure supplement 3a and Figure 3—figure supplement 1b). This can be better appreciated in the Figure 1-Video 1, 3 and 4. The magnified images focusing on the Golgi membrane is now shown in Figure 1b. There are few small “dots” that sometimes appear with overexpression of these biosensors. We believe that these are likely to be biosensor protein aggregates. These dots do not change with agonist addition or receptor activation. Thus, we don’t think they are a distinct membrane compartment.

In the document given for review, the images seem blurry. Were they image processed, e.g. smoothened?

We have provided the original RBG files. These images are neither processed nor smoothened. We apologize that this reviewer had poor-quality figures. It is possible that image quality may have degraded in case there were conversions between Mac and PC once we had submitted the images. We have provided a PDF version of these figures in the hope that image quality may be better preserved.

Are the normalized fluorescence traces for HEK293 in Figure 1 and 2 the same data shown twice? Ideally these should be controls paired to the experimental conditions.

We now provide a new figure (Figure 1d) in which the data are all shown in one graph. Please note that the new data is based on new repeats.